# Spectral Collapse Drives Loss of Plasticity in Deep Continual Learning

## Abstract

We investigate why deep neural networks suffer from *loss of plasticity* in deep continual learning, failing to learn new tasks without reinitializing parameters. We show that this failure is preceded by Hessian spectral collapse at new-task initialization, where meaningful curvature directions vanish and gradient descent becomes ineffective. To characterize the necessary condition for successful training, we introduce the notion of $\tau$-trainability and show that current plasticity preserving algorithms can be unified under this framework. Targeting spectral collapse directly, we then discuss the Kronecker factored approximation of the Hessian, which motivates two regularization enhancements: maintaining high effective feature rank and applying $L2$ penalties. Experiments on continual supervised and reinforcement learning tasks confirm that combining these two regularizers effectively preserves plasticity.

## 1 Introduction

Human intelligence is marked not by mastering a single task, but by the ability to accumulate knowledge, refine it, and adapt to new challenges. Likewise, a central aspiration of artificial intelligence is to create systems that can learn and adapt throughout their lifetimes. Achieving this capacity in artificial systems would ensure that agents that master preconceived benchmarks remain useful, relevant, and robust in the open world (Javed & Sutton, 2024).

Deep learning and deep reinforcement learning have demonstrated remarkable progress, reaching human and even superhuman performance in image recognition (Siméoni et al., 2025), game playing (Silver et al., 2017), and reasoning (Bakhtin et al., 2022). But these advances have largely been realized under stationary data distributions. **Continual learning** concerns a system's ability to perform well through a sequence of evolving tasks. Within this framework, two complementary objectives are often distinguished: (i) *maintaining plasticity*, meaning the capacity to acquire new knowledge (Dohare et al., 2024), and (ii) *preventing catastrophic forgetting*, where the learner forgets how to solve tasks it previously mastered (Parisi et al., 2018; Kirkpatrick et al., 2017).

Our work focuses on the first objective. Despite significant empirical progress in the area (Lyle et al., 2023; 2024; Abbas et al., 2023), there remains no unifying consensus on what drives plasticity loss (Klein et al., 2024). Several disparate explanations have been investigated, including inactive neurons which output constants regardless of the inputs (Bjorck et al., 2021; Sokar et al., 2023), the reduction in feature expressiveness associated with large parameter norms (Lyle et al., 2024), and overfitting (Igl et al., 2020). Recent work has begun to connect plasticity loss to second-order properties, such as the stable rank of the Hessian or layer-wise spectral conditioning (Lewandowski et al., 2024b;a). Building on these insights, we argue these seemingly disparate phenomena are in fact different facets of the same phenomenon, which we call **spectral collapse**: a degeneration of the Hessian eigenspectrum, indicating loss of curvature in most directions.

To illustrate the importance of second order information in continual learning, we consider a toy problem that involves sequentially optimizing two functions $f_1, f_2 : \mathbb{R}^2 \to \mathbb{R}$ in Figure 1. Each function has one bowl-shaped minimum with rich curvature information, and a second valley-shaped minimum with curvature in mostly one direction. We obtain $f_2$ as a fixed linear reparameterization of $f_1$, modeling a task switch. In the first task, standard gradient descent (GD) finds the valley, while curvature-regularized GD finds the bowl. After switching to the second task, GD, now initialized in the valley region, exhibits symptoms of ill-conditioning (zig-zagging and

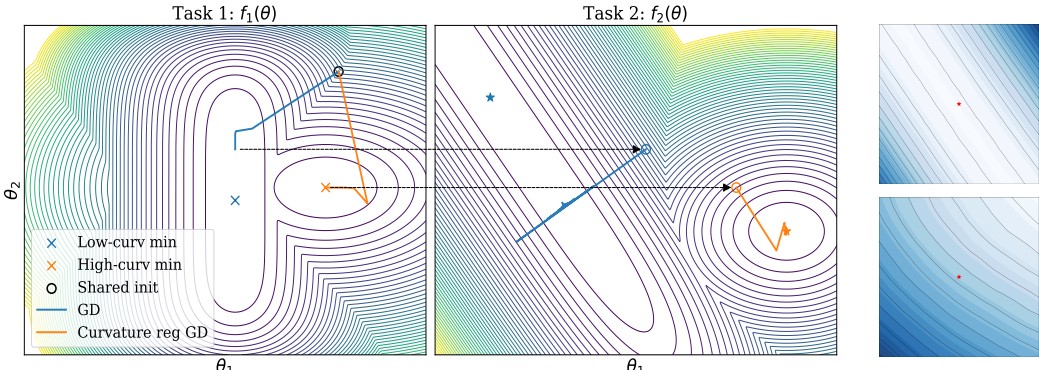

Figure 1: (Left) An illustration of gradient descent (GD) versus curvature-regularized GD in a toy two-minima setting. Both algorithms are initialized from the same point in Task 1 and run for the same number of steps. The curvature regularized method finds high-curvature minimum and remains well conditioned on Task 2, while GD is initialized at an ill-conditioned subspace on Task 2 and zig-zags. Full details can be found in Appendix C. (Right) 2D loss landscape slices projected onto the top two principal directions. Top right: Standard backpropagation. Bottom right: L2-ER. Red dots denote the current parameters.

slow progress), whereas curvature-regularized GD converges rapidly. We show that our proposed method, L2-ER, provides one such practical instantiation of this idea. The 2D loss-landscape on Continual ImageNet at Task 800 initialization closely mirror this toy example: standard backpropagation (BP) finds an elongated, flat valley, whereas $L2$-ER, our curvature-preserving algorithm, maintains a well-conditioned, bowl-shaped landscape (see Fig. 1 and Appendix A for additional views). Indeed, this toy example reflects what we observe at scale: spectral collapse, and the resulting low-curvature initialization, is strongly tied to poor task-level training performance and a progressive loss of plasticity.

These insights motivate the central themes of this work. Building on them, we make three key contributions. First, through an extensive empirical study of the Hessian eigenspectrum across three continual learning tasks, multiple architectures, and a range of algorithms, we identify spectral collapse as a consistent driver of loss of plasticity. Second, we define $\tau$-trainability as a framework for analyzing successful training in continual learning, and show that the number of inactive neurons upper bounds $\tau$-trainability via their effect on the Hessian rank. Finally, motivated by this theoretical perspective, we introduce $L2$-Effective Feature Rank ($L2$-ER), a simple regularizer that stabilizes the Hessian spectrum. Across diverse benchmarks, $L2$-ER prevents spectral collapse and maintains plasticity[1].

## 2 RELATED WORK

**Causes of Plasticity Loss** Several pathologies have been proposed to explain loss of plasticity, in both continual supervised and reinforcement learning (Lyle et al., 2023; Klein et al., 2024). For example, **dormant neurons**, (which output a positive value on all inputs), reduce a network's expressive power, while **dead neurons** (output zero on all inputs) prevent gradient flow (Montufar et al., 2014; Sokar et al., 2023). Reductions in **effective feature rank**, which measures the intrinsic dimensionality of feature representations and directly relates to the persistence of dead neurons, has also been linked to loss of plasticity (Roy & Vetterli, 2007; Dohare et al., 2024), as has **parameter norm growth**, which leads to ill-conditioned Hessians that cause numerical instability (Obando-Ceron et al., 2024; Lyle et al., 2024). Lyle et al. (2024) propose a "swiss cheese" model that attempts to preserve plasticity by targeting all these pathologies simultaneously.

**Mitigating Plasticity Loss** Existing methods for mitigating loss of plasticity are targeting one or more of the aforementioned pathologies (Klein et al., 2024). Broadly speaking, these approaches fall into two categories: (i) continuous interventions, which apply at every step of the optimization,

---

[1]We release an open-source JAX implementation and benchmark to support future research.

and (ii) intermittent interventions, which periodically reset or perturb parameters (Juliani & Ash, 2024). For example, Lyle et al. (2024) shows that combining $L2$ regularization (to control parameter norm growth) with layer normalization (to stabilize pre-activation distributions) is highly effective. Other approaches regularize the singular values of the layer-wise Jacobians (Lewandowski et al., 2024a) to maintain parameter distributions close to a random initialization (He et al., 2015; Hinton & Salakhutdinov, 2006). Architectural changes such as PELU (Godfrey, 2019), CRELU (Shang et al., 2016; Abbas et al., 2023), and adaptive rational activations (Delfosse et al., 2021) avoid the saturation of neurons. In contrast, periodic interventions, such as Shrink & Perturb (Ash & Adams, 2020), continual backpropagation (CBP) (Dohare et al., 2024), and ReDo (Sokar et al., 2023), selectively reset dormant neurons. Empirically, hybrid strategies—typically involving $L2$ regularization combined with either architectural or periodic interventions—exhibit the strongest performance (Lyle et al., 2024; Juliani & Ash, 2024), implying that plasticity loss can emerge from multiple mechanisms.

**Curvature Based Methods** Our work contributes to a recent line of inquiry that connects loss of plasticity to the second-order properties of neural networks. Prior work by Lewandowski et al. (2024b) provides strong empirical evidence that loss of plasticity correlates with a decline in the stable rank of a partial Hessian, proposing a Wasserstein penalty to maintain characteristics of the initial parameter distribution. In a complementary approach, Lewandowski et al. (2024a) focus on the spectral properties of the layer-wise weight matrices rather than the Hessian, using spectral regularization to keep the largest singular value near one, thereby maintaining favorable conditioning and gradient diversity. Our contributions advances this frontier in several key ways. Similarly, Parseval regularization (Chung et al., 2024) improves optimization by enforcing orthogonality in the weight matrices to preserve gradient norms. First, where prior work only shows correlation, we develop a more formal, curvature-centric theory of trainability based on the full Hessian rank— a direct measure of the number of actionable optimization directions. Second, our proposed $L2$-ER regularizer is derived directly from this theory; by maximizing effective feature rank Kumar et al. (2020) with a calibrated $L2$-penalty, it provides a principled mechanism for provably increasing the rank of the Hessian's Gauss-Newton approximation. Finally, our theoretical framework is unifying: it provides a lens through which disparate plasticity-preserving techniques, like continual backpropagation (Dohare et al., 2024) and Shrink & Perturb (Ash & Adams, 2020), can all be understood as mechanisms that prevent spectral collapse.

## 3 PRELIMINARIES

**Deep Neural Networks** A multi-layered-perceptron (MLP), also called a feedforward neural network, is constructed by chaining together $L$ functions, i.e., $F_L \circ \cdots \circ F_1$, one per layer. Given an input (resp. output) dimension of $M_{l-1}$ (resp. $M_l$), the function at layer $l$, for $l \in \{1, \ldots, L\}$, is a map $F_l : \mathbb{R}^{M_{l-1}} \to \mathbb{R}^{M_l}$. These dimensions correspond to the number of neurons. The total number of hidden neurons is $M \doteq \sum_{l=1}^{L-1} M_l$. The function $F_l$ at layer $l$, again for $l \in \{1, \ldots, L\}$, is the composition of an element-wise non-linear function $\sigma_l : \mathbb{R} \to \mathbb{R}$ and a weight matrix $\mathbf{W}_l \in \mathbb{R}^{M_l \times M_{l-1}}$, so that $F_l = \sigma_l \circ \mathbf{W}_l$. Our non-linearity of choice is the ReLU activation, defined as $\max\{0, x\}$. We collect all parameters into a single vector using the vec operator, so that the MLP parameterization is represented by $\boldsymbol{\theta} \doteq \text{vec}(\mathbf{W}_1, \ldots, \mathbf{W}_L) \in \mathbb{R}^P$, where $P = \sum_{l=1}^{L} M_l M_{l-1}$ is the total parameter count. Denoting the input (resp. output) dimension of the MLP by $I \doteq M_0$ (resp. $O \doteq M_L$), we can now express the MLP map as $F_{\boldsymbol{\theta}} : \mathbb{R}^I \to \mathbb{R}^O$. A task $\tau$ is described by a loss function $\ell_\tau$ and a data distribution $\boldsymbol{d}_\tau$. Given such a task $\tau$, a learning algorithm seeks neural network's parameter values that minimize its expected loss $\mathcal{L}_\tau(\boldsymbol{\theta}) \doteq \mathbb{E}_{\boldsymbol{d}_\tau}[\ell_\tau(F_{\boldsymbol{\theta}}(\boldsymbol{x}), \boldsymbol{y})]$ on the task. The **Jacobian** is defined as the vector of first derivatives of the loss function w.r.t. the network parameters, i.e., $J \doteq \nabla_{\boldsymbol{\theta}} \ell_\tau \in \mathbb{R}^P$, while the main object of our investigation, the **Hessian**, is the matrix of second derivatives, i.e., $\mathbf{H} \doteq \nabla_{\boldsymbol{\theta}}^2 \ell_\tau \in \mathbb{R}^{P \times P}$. Together, the Jacobian and the Hessian capture up to second-order changes in the loss, i.e., when the parameters change by a vector $\delta \in \mathbb{R}^P$, the loss can be approximated as: $\mathcal{L}_\tau(\boldsymbol{\theta} + \delta) \approx \mathcal{L}_\tau(\boldsymbol{\theta}) + \nabla_{\boldsymbol{\theta}} \mathcal{L}_\tau^\top \delta + \frac{1}{2}\delta^\top \nabla_{\boldsymbol{\theta}}^2 \mathcal{L}_\tau \delta$.

**The Hessian** The Hessian offers valuable insight into the loss landscape of a neural network (Pouplin et al., 2023). For example, one line of research is concerned with large connected regions in weight space where the error remains approximately constant (Hochreiter & Schmidhuber, 1997);

such regions may be less prone to overfitting and result in smaller generalization gaps (Cha et al., 2021). Since the Hessian $\mathbf{H}$ is symmetric, there exists an orthonormal basis of eigenvectors $\{\mathbf{v}_i\}$ and real eigenvalues $\{\lambda_i\}$ satisfying $\mathbf{H}\mathbf{v}_i = \lambda_i \mathbf{v}_i$, for all $i \in \{1, \ldots, P\}$. Each eigenvalue $\lambda_i$ measures the second-order curvature of the loss in the direction of the eigenvector $\mathbf{v}_i$. A large eigenvalue indicates a steep change in the loss in the direction of the corresponding eigenvector, while a small eigenvalue indicates a flat change. Given the importance of the eigenvalues of the Hessian, the distribution of all eigenvalues, known as the **eigenspectrum**, can provide even richer insights into a neural network's loss landscape.

## 4 Continual Learning

We define the continual learning problem is over a sequence of $T$ tasks and an agent, represented by a neural network with parameters $\boldsymbol{\theta} \in \mathbb{R}^m$. Each task comes equipped with an evaluation metric $f_\tau : \mathbb{R}^m \to \mathbb{R}$, such as accuracy for supervised learning or expected returns for reinforcement learning.

Formally, we define the continual learning problem as finding a sequence of parameters $\{\boldsymbol{\theta}_\tau^*\}_{\tau=0}^{T-1}$ s.t. $\boldsymbol{\theta}_\tau^*$ maximizes the evaluation metric $f_\tau(\boldsymbol{\theta}_\tau)$ for task $\tau$: i.e., find $\boldsymbol{\theta}_\tau^* \in \arg\max_{\boldsymbol{\theta}_\tau} f_\tau(\boldsymbol{\theta}_\tau)$, for all tasks $\tau \in \{0, \ldots, T-1\}$.

Since these evaluation metrics may not be directly optimizable (e.g., they may be non-differentiable), we further assume a (possibly regularized) surrogate expected loss function $\mathcal{L}_\tau : \mathbb{R}^m \to \mathbb{R}$, for each task $\tau$, which is intended to maximize the evaluation metric, but is generally better behaved. Additionally, we assume the learner has a budget of $K$ learning steps per task.

A solution to the continual learning problem is generated by an algorithm $\mathcal{A} : \mathbb{R}^m \to \mathbb{R}^m$, which takes as input a function of the parameters learned during the previous task (e.g., the parameters for task $\tau$ are initialized to be the learned parameters of task $\tau - 1$: i.e., $\boldsymbol{\theta}_\tau^{(0)} \doteq \boldsymbol{\theta}_{\tau-1}^{(K)}$) and outputs the parameters it learns for the current task $\tau$, by optimizing $\tau$'s surrogate loss function $\mathcal{L}_\tau$.

## 5 Plasticity Through the Lens of $\tau$-Trainability

Dead neurons are known to impede a network's ability to learn, by limiting expressive capacity (Maas et al., 2013). Accordingly, prior work (Shin & Karniadakis, 2020) ties the number of active neurons at initialization to successful training.

**Definition 5.1** (Successful Training). *Given a continual learning problem, and an small error tolerance $\upsilon$, we write $\mathbb{I}(f_\tau(\boldsymbol{\theta}_\tau^{(K)}) \geq \sup_{\boldsymbol{\theta}_\tau} f_\tau(\boldsymbol{\theta}_\tau) - \upsilon)$ to indicate successful training on task $\tau$. Moreover, we say a neural network is plastic if this condition holds, for all tasks $\tau \in [T]$.*

In (Shin & Karniadakis, 2020), a network is defined to be trainable if it is initialized with at least $a_\tau$ active neurons, where $a_\tau$ is task dependent. A network being trainable is a necessary condition for successful training. Among other pathologies, strong gradient collinearity can reduce functional expressivity and impede learning, as argued by Lyle et al. (2023) and evidenced in Section 8.

We generalize this notion of trainability to continual learning. Like Shin & Karniadakis (2020), we rely on a task and optimizer-dependent threshold, but our threshold relates to the neural network curvature. Curvature is known to be an important factor in successful training. For instance, Du et al. (2018) show that, under overparametrization, a linear convergence can be proved when the empirical Gram matrix has its minimum eigenvalue bounded below by a positive constant.

**Definition 5.2** ($\tau$-Trainable). *A neural network is said to be $\tau$-trainable if its Hessian rank at task $\tau$ initialization is at least $\rho_\tau$, a task and optimizer-dependent threshold. Let $T_\tau$ denote the event that a neural network is $\tau$-trainable. The probability of a neural network being $\tau$-trainable is referred to as $\tau$-trainability.*

We say a Hessian suffers from spectral collapse when the majority of its eigenvalues cluster around zero at a task initialization. To more precisely quantify spectral collapse, we define $\epsilon\text{-rank}(\mathbf{H}) \doteq \#\{i \mid |\lambda_i(\mathbf{H})| > \epsilon\}$, the number of Hessian's eigenvalues whose absolute values exceed a threshold $\epsilon > 0$. The $\epsilon\text{-rank}(\mathbf{H})$ indicates the number of directions with non-negligible curvature, while the

eigenvalues' magnitudes quantify how much curvature exists in those directions. When spectral collapse occurs, the optimization landscape becomes *effectively flat in most directions*, rendering gradient-based updates ineffective. We demonstrate, in Figure 2 (left), a strong association between the $\epsilon$-Hessian rank and classification accuracy on Continual ImageNet. For example, a normalized $\epsilon$-Hessian rank of at least 0.5 is a necessary condition for achieving $\sim 80\%$ classification accuracy on Continual ImageNet, further motivating definition 5.2.

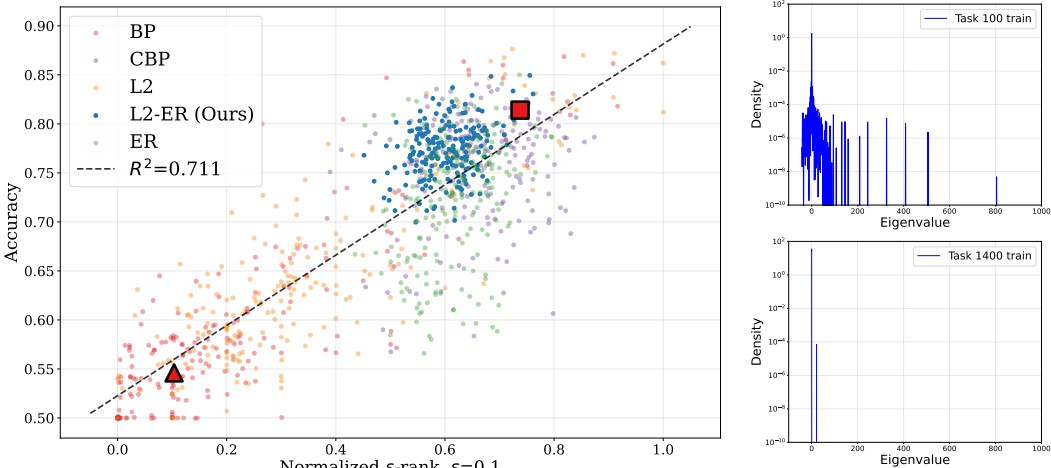

Figure 2: (Left) Accuracy (i.e., successful training) vs. Curvature (i.e., normalized $\epsilon$-rank($\mathbf{H}$)) [2] on Continual ImageNet. A linear fit (dotted line) highlights the positive association between curvature and accuracy. (Top Right) ■: BP Hessian eigenspectrum *before spectral collapse* at task 100. (Bottom Right) ▲: BP Hessian eigenspectrum *after spectral collapse* at task 1400.

On the contrary, a low $\epsilon$-rank($\mathbf{H}$) representing spectral collapse is strongly related to unsuccessful training. We run extensive experiments across diverse architectures and settings (see Section 8) to support the claim that *preventing spectral collapse is necessary for successful training in continual learning*. The detailed results including the evolution of Hessian eigenspectrum, are presented in Appendices G to J, along with experiments on Permuted MNIST and Incremental CIFAR.

## 6 Dead Neurons Upper Bound $\tau$-trainability

This section examines the dynamics of dead neurons across task shifts in continual learning. We first formalize the concept of dead neurons and then, in Theorem 6.2, we establish that their count imposes an upper bound on the rank of the Hessian. We subsequently characterize the probability that a neuron remains dead following a task shift, which allows us to derive a probabilistic bound on the network's $\tau$-trainability. These results provide a unifying framework, showing that various neuron-resetting approaches can be understood as mechanisms that aim to preserve the Hessian rank required for $\tau$-trainability. Proofs of all our theoretical results can be found in Appendix B.

Throughout this section, we study a two-layer MLP with ReLU activations and parameters $\boldsymbol{\theta} = \{\mathbf{W}_1, \mathbf{b}_1, \mathbf{W}_2, \mathbf{b}_2\}$ with $\mathbf{W}_1 \in \mathbb{R}^{H \times I}$, $\mathbf{b}_1 \in \mathbb{R}^H$, $\mathbf{W}_2 \in \mathbb{R}^{O \times H}$, $\mathbf{b}_2 \in \mathbb{R}^O$, where $I$ is the input dimension, $H$ is the hidden dimension, and $O$ is the output dimension. Let $\mathcal{X}_\tau \subset \mathbb{R}^I$ be the domain of training inputs for task $\tau$. We assume normalized input data, so that for all $\tau \in [T]$, $\mathcal{X}_\tau \subset \mathcal{B}_r(0)$. We denote unit $j$ in layer $l$ by $\mathbf{W}_{[j,:],l}$.

**Definition 6.1** ($\tau$-Dead Neurons). *We say a hidden unit $j$ in layer $l$ is $\tau$-dead when its output is 0 after activation i.e., $\sigma(\mathbf{W}_{[j,:],l}^\top F_{l-1}(\boldsymbol{x}) + \mathbf{b}) = 0$, for all $\boldsymbol{x} \in \mathcal{X}_\tau$. Furthermore, given $\gamma \geq 0$, we say neuron $j$ is $\gamma$-margin dead for task $\tau$ if $\mathbf{W}_{[j,:],l}^\top F_{l-1}(\boldsymbol{x}) + \mathbf{b} \leq -\gamma$, for all $\boldsymbol{x} \in \mathcal{X}_\tau$.*

The following theorem upper bounds the Hessian rank of a task $\tau$, and in turn, $\tau$-trainability, in terms of the number of dead neurons at its initialization.

---

[2]The $x$-axis is **normalized $\epsilon$-rank(H)**, i.e., $\epsilon$-rank($\mathbf{H}$)$/\#\{$eigenvalues of $\mathbf{H}\}$, for $\epsilon = 0.1$.

**Theorem 6.2.** *Let $P = H(I + 1) + O(H + 1)$ be the full parameter dimension, and let $k_\tau$ be the number of dead neurons on $\mathcal{X}_\tau$ at task $\tau$'s initialization. The Hessian rank satisfies $\text{rank}(\mathbf{H}_\tau^{(0)}) \leq P - k_\tau(I + O + 1)$. Equivalently, the number of dead neurons satisfies $k_\tau \leq \frac{P - \text{rank}(\mathbf{H}_\tau^{(0)})}{I + O + 1}$. Moreover,*

$$\mathbf{Pr}(T_\tau = 1) \leq \mathbf{Pr}\left( k_\tau \leq \frac{P - \rho_\tau}{I + O + 1} \right). \tag{1}$$

**Dead Neurons Across Tasks** Dead neurons need not persist across tasks, as a previously constant output value may vary with new data. Previous work on trainability (Shin & Karniadakis, 2020) has explored the probability of a neuron being born dead under different initialization schemes. In our setup, the learner's parameters for task $\tau$ are initialized as the learned parameters on task $\tau - 1$. We study dead neurons across pairs of consecutive tasks by measuring the worst-case shift in inputs, which we call a **task shift**. Let the Hausdorff distance between two training sets be defined as $\Delta_\tau \doteq \max\{\sup_{\boldsymbol{x} \in \mathcal{X}_\tau} \inf_{\boldsymbol{x}' \in \mathcal{X}_{\tau'}} \|\boldsymbol{x} - \boldsymbol{x}'\|, \sup_{\boldsymbol{x}' \in \mathcal{X}_{\tau'}} \inf_{\boldsymbol{x} \in \mathcal{X}_\tau} \|\boldsymbol{x}' - \boldsymbol{x}\|\}$.

**Lemma 6.3.** *Let $\mathcal{X}_\tau$ and $\mathcal{X}_{\tau'}$ be input datasets for two consecutive tasks. If neuron $j$ is $\gamma$-margin dead on $\mathcal{X}_\tau$, then neuron $j$ remains dead on $\mathcal{X}_{\tau'}$, if $\gamma_\tau^j \geq \Delta_\tau \|\mathbf{W}_{[j,:],1}\|$.*

**Definition 6.4.** *If a neuron $j$ is dead at the end of task $\tau$, we define $\xi_\tau^j$ as the indicator function that neuron $j$ remains dead throughout task $\tau'$: i.e., $\xi_\tau^j = \mathbf{1}\left\{\gamma_\tau^j \geq \Delta_\tau \|\mathbf{W}_{[j,:],1}\|\right\}$. Furthermore, we let $p_\tau^j \doteq \mathbf{Pr}(\xi_\tau^j = 1)$ denote its persistence probability.*

If $\gamma_\tau^j$ and $\mathbf{W}_{[j,:]}$ are constants that result from training a neural network on task $\tau$, then $\xi_\tau^j$ is a random variable on the task shifts $\Delta_\tau$. Our next result, Theorem 6.5 states, if the number of dead neurons is large at the end of task $\tau$ and the probability of the dead neurons staying dead is large from task $\tau$ to $\tau'$, then the probability of task $\tau'$ being trainable is small.

**Theorem 6.5.** *Consider a 2-layer ReLU network with $n$ neurons and hidden layer dimension $H$ and a sequence of tasks $\tau \in \{1, \ldots, T\}$. At the end of task $\tau$, define $D_\tau$ to be the set of $\tau$-dead neurons (i.e., $|D_\tau| = k_\tau$). Let $m_{\tau'} = \frac{P - \rho_{\tau'}}{I + O + 1}$ be the maximum allowable number of dead neurons so that task $\tau'$ is trainable. If $\sum_j p_\tau^j \geq H - m_{\tau'}$, then the one-sided Chebyshev (Cantelli) inequality yields the following upper bound on $\tau'$-trainability:*

$$\mathbf{Pr}(T_{\tau'} = 1) \leq \mathbf{Pr}\left( \sum_j \xi_\tau^j \leq n - m_{\tau'} \right) \leq \frac{\mathbf{Var}(\sum_j \xi_\tau^j)}{\left[\sum_j p_\tau^j - (n - m_{\tau'})\right]^2}. \tag{2}$$

**Remark 6.6.** *Theorem 6.5 bounds the probability of $\tau$-trainability via the sum $S \doteq \sum_j \xi_\tau^j$ of the indicator variables that neuron $j$ remains dead at the end of task $\tau$. The bound is tighter when $S$ is large and the $\xi_\tau^j$'s are (approximately) independent and $\sum_j p_\tau^j - (n - m_{\tau'})$ is small, or when the variance of $S$ is small —conditions under which the Cantelli inequality closely tracks the tail. Moreover, Lemma B.1 gives a lower bound $p_\tau^j \geq 1 - 2 \exp\left( \log N - N \left( \frac{\gamma_\tau^j}{\|\mathbf{W}_{[j,:],1}\| \cdot r} \right)^I \right)$, if each task dataset consists of $N$ points drawn i.i.d. from the uniform distribution on $\mathcal{B}_r(0)$, which implies $p_\tau^j$ grows with the sample size $N$, and shrinks with the input dimension $I$. In particular, in low–intrinsic-dimension regimes (e.g., images), if the number of dead neurons is large at the end of task $\tau$, then $\sum_j p_\tau^j$ is large and the probability of task $\tau'$ being trainable is small.*

**A Unified Framework** Our analysis formalizes the mechanism behind plasticity loss: when many neurons are dead and stay dead across tasks, the expected number of live neurons falls below what is needed to satisfy the next task's minimal curvature threshold $\rho_{\tau'}$ required for $\tau$-trainability. Similarly, it has been found that growing parameter norms may lead to gradients with sparse and/or co-linear gradient covariance matrices, which in turn results in a low rank Hessian (Obando-Ceron et al., 2024; Lyle et al., 2023). Our characterization of $\gamma$-dead neurons also explains why dead neurons can unfreeze after a suitably large task-shift (Lyle et al., 2024), which explains why low-dimension regimes, like images, are unlikely to revive these neurons. Our analysis also

unifies revival methods like continual backpropagation (Dohare et al., 2024) and curvature-based regularization (Lewandowski et al., 2024a) under a single mechanism. Revival operations act by mechanistically reducing probabilities $p_\tau^j$ of dead neurons. Curvature regularizers act by distributing curvature to prevent concentration in a few directions, effectively raising the Hessian rank. Both interventions aim to maximize $\tau$-trainability, either by lowering $\sum_j \xi_\tau^j$ or by increasing the attainable $\text{rank}(\mathbf{H}_\tau^0)$. This equivalence explains why seemingly disparate techniques improve plasticity in continual learning: they target the same limiting factor, the availability of independent curvature directions.

# 7 MITIGATING SPECTRAL COLLAPSE

To maximize $\tau$-trainability, we introduce a loss regularizer that directly targets the approximate Hessian rank. MLPs can have thousands or even millions of parameters meaning their Hessian is a matrix with elements in the range of millions to trillions. Therefore, we need a method to influence the Hessian rank through computable proxies. To do so, we first obtain the Gauss-Newton decomposition of the Hessian as follows (Zhao et al., 2024):

$$\nabla_{\boldsymbol{\theta}}^2 \ell = \mathbf{H} = \underbrace{\frac{1}{N} \sum_{n=1}^N J_n^\top \left[ \nabla_{F_{\boldsymbol{\theta}}}^2 \ell(F_{\boldsymbol{\theta}}(\boldsymbol{x}_n), \boldsymbol{y}_n) \right] J_n}_{\mathbf{H}_{GGN}} + \underbrace{\frac{1}{N} \sum_{n=1}^N \sum_{o=1}^O \left[ \nabla_{F_{\boldsymbol{\theta}}} \ell(F_{\boldsymbol{\theta}}(\boldsymbol{x}_n), \boldsymbol{y}_n) \right]_o \nabla_{\boldsymbol{\theta}}^2 F_{\boldsymbol{\theta}}^o(\boldsymbol{x}_n)}_{\mathbf{R}},$$

(3)

where at the $n$th sample, $J_n \in \mathbb{R}^{O \times P}$ is the Jacobian of the models with respect to the parameters, $\left[ \nabla_{F_{\boldsymbol{\theta}}} \ell_n \right]_o$ is $\mathbb{R}^{O \times 1}$ vector which is the Jacobian of the loss with respect to the logits of the model, $[\cdot]_o$ is the scalar at index $o$, $\nabla_{F_{\boldsymbol{\theta}}}^2 \ell \in \mathbb{R}^{O \times O}$ is the Hessian of the loss with respect to the logits, and $\nabla_{\boldsymbol{\theta}}^2 F_{\boldsymbol{\theta}}^o(\boldsymbol{x}_n) \in \mathbb{R}^{P \times P}$ is the Hessian matrix of the $o$-th output element with respect to the model parameters. Simply dropping the residual term $\mathbf{R}$, yields the Generalized Gauss-Newton (GGN) matrix $\mathbf{H}_{GGN}$, a faithful proxy for the Hessian (Dangel et al., 2025).

A neural network has parameters corresponding to each layer. Given a specific parameter, $\mathbf{W}_{[j,i],l}$ in layer $l$ and another parameter, $\mathbf{W}_{[j',i'],l'}$, in a different layer (i.e., $l \neq l'$), in practice $\partial^2 \mathcal{L} / \partial \mathbf{W}_{[j,i],l} \partial \mathbf{W}_{[j',i'],l'}$ is approximately 0 in (Becker et al., 1988), where $\mathcal{L}$ is the loss. Therefore, Hessian admits a natural "layer-wise block" structure. Kronecker-factored approximate curvature (KFAC) is a method to approximate a matrix using only block-diagonal $\{M_l\}_{l \in L}$ elements. In particular, KFAC is frequently used for approximating the numerical quantities related to the Hessian like the $\mathbf{H}_{GGN}$ or Fisher Information matrices (Dangel et al., 2025; Martens, 2016). We therefore provide a justification for our effective rank based regularizer using a KFAC approximation of the Hessian. We then argue that increasing the rank of the KFAC-approximate $\mathbf{H}_{GGN}$ implies increasing the rank of the true Hessian for mean-square-error (MSE) and softmax-cross-entropy (SCE) losses.

As an analogue of a multi-layer MLP, consider a multi-layer linear neural network without bias, which we employ to justify the regularizer. The layer $l$ in a neural network has an associated weight matrix $\mathbf{W}_l$. For a dataset of size $N$, let $\mathbf{a}_l^n$ be the input activations to the layer for the $n$th datum, producing pre-activation feature $\boldsymbol{z}_l^n = \mathbf{W}_l \mathbf{a}_l^n$. Let $\boldsymbol{g}_l^{n,o} = \nabla_{\boldsymbol{z}_l^n} \ell(\cdot)$ be the gradient of the loss with respect to the layer's output. Now, using the properties of the Kroneker product, it is possible to define $\mathbf{H}_{GGN}$ as a sum of Kroneker products:

$$\mathbf{H}_{GGN} = A \sum_{n=1}^N \sum_{o=1}^O (\mathbf{a}_l^n \mathbf{a}_l^{n\top}) \otimes (\boldsymbol{g}_l^{n,o} \boldsymbol{g}_l^{n,o\top}),$$

(4)

where $A$ is an accumulation factor[3] and $\otimes$ is the Kronecker product. The term $\mathbf{a}_l^n \mathbf{a}_l^{n\top}$ captures the input feature statistics, while $\boldsymbol{g}_l^{n,o} \boldsymbol{g}_l^{n,o\top}$ captures the backpropagated output gradient statistics. The sum of Kronecker products is *still* computationally intractable. However, we can now apply the KFAC approximation by decoupling the summations over the inputs and gradients (Dangel et al., 2025):

$$\mathbf{H}_{GGN} \approx \widehat{\mathbf{H}}_{GGN} = \left( A \sum_{n=1}^N \mathbf{a}_l^n \mathbf{a}_l^{n\top} \right) \otimes \left( \frac{1}{N} \sum_{n=1}^N \sum_{o=1}^O \boldsymbol{g}_l^{n,o} \boldsymbol{g}_l^{n,o\top} \right), \tag{5}$$

In our sample based setting, the approximation error can be expressed as the difference between the average values of the Kronecker products and the Kronecker product of average values:

$$\text{KFAC Error} = \mid \mathbf{H}_{GGN} - \widehat{\mathbf{H}}_{GGN} \mid \tag{6}$$

We explain the error derivation further in Appendix D. The approximation error is small when the joint distribution over $\mathbf{a}_l, \mathbf{a}_l^\top, \boldsymbol{g}_l$, and $\boldsymbol{g}_l^\top$ is a multivariate Gaussian, which appears to hold in practice (Martens & Grosse, 2015). Alternatively, the approximation can be viewed as an assumption of statistical independence between $\mathbf{a}_n^l \mathbf{a}_n$ and $\boldsymbol{g}_n^l \boldsymbol{g}_n^\top$, which holds for deep linear networks with squared-error loss (Bernacchia et al., 2018).

**Maximizing the Rank of the Input Covariance Matrix** For each layer $l$, the KFAC approximation $\widehat{\mathbf{H}}_{GGN}$ has the block form $\mathbb{E}_n[\mathbf{a}_l^n \mathbf{a}_l^{n\top}] \otimes \mathbb{E}_n[\boldsymbol{g}_l^n \boldsymbol{g}_l^{n\top}]$. Since $\text{rank}(\boldsymbol{A} \otimes \boldsymbol{B}) = \text{rank}\boldsymbol{A} \cdot \text{rank}\boldsymbol{B}$, the rank of $\widehat{\mathbf{H}}_{GGN}$ is monotone in both the rank of the input covariance, $\mathbb{E}_n[\mathbf{a}_l^n \mathbf{a}_l^{n\top}]$ and the gradient covariance $\mathbb{E}_n[\boldsymbol{g}_l^n \boldsymbol{g}_l^{n\top}]$. The rank of the input covariance matrix $\mathbb{E}_n[\mathbf{a}_l^n \mathbf{a}_l^{n\top}]$ coincides with rank of the input representation of layer $l$ itself. This latter quantity, called the **effective feature rank (ER)**, has been shown to be a useful indicator of plasticity (Dohare et al., 2024; Lyle et al., 2023), and single task RL (Kumar et al., 2020).

**Maximizing the Rank of the Hessian** Adding $L2$ regularization is required to ensure the rank of the Hessian is sufficiently high, since acting on $\mathbf{H}_{GGN}$ does not take into account the contribution of the residual $\mathbf{R}$. In Theorem B.3, we show that there exists a suitable $L2$ regularization parameter to guarantee that increasing $\text{rank}(\mathbf{H}_{GGN})$ implies increasing $\text{rank}(\mathbf{H})$. Then, the rank property of Kronecker products implies that $\sum_{l \in L}(\text{rank}(\mathbb{E}_n[\mathbf{a}_l^n \mathbf{a}_l^{n\top}])\text{rank}(\mathbb{E}_n[\boldsymbol{g}_l^n \boldsymbol{g}_l^{n\top}])) \geq \rho_\tau$ is a sufficient condition for task $\tau$ to be $\tau$-trainable.

**L2-ER Regularization** Given task $\tau$, with data distribution $\boldsymbol{d}_\tau$ and loss function $\ell_\tau$, standard gradient descent minimizes $\mathbb{E}_{\boldsymbol{d}_\tau}[\ell_\tau(F_{\boldsymbol{\theta}}(\boldsymbol{x}), \boldsymbol{y})]$. We propose the $L2$-ER regularizer, which combines effective rank and $L2$ penalties, leading to a theoretically grounded and practical objective:

$$\min_{\boldsymbol{\theta}} \mathcal{L}_\tau(\boldsymbol{\theta}) = \min_{\boldsymbol{\theta}} \underbrace{\mathbb{E}_{\boldsymbol{d}_\tau}\left[ \ell_\tau(F_{\boldsymbol{\theta}}(\boldsymbol{x}), \boldsymbol{y}) \right]}_{\text{Loss}} - \underbrace{\text{erank}\left( \sum_{l \in L} \mathbb{E}_n\left[ \mathbf{a}_l^n \, \mathbf{a}_l^{n\top} \right] \right)}_{\text{Effective rank penalty}} + \underbrace{\lambda \|\boldsymbol{\theta}\|_2^2}_{L2 \text{ regularization}}$$

In summary, we add an empirically computable effective rank penalty to the objective that increases the $\epsilon$-rank of the approximate GGN Hessian, $\widehat{\mathbf{H}}_{GGN}$, which in turn effects the bulk of the eigenspectrum of the true Hessian.

## 8    EXPERIMENTS

We conduct experiments across a range of continual learning tasks adapted from supervised and reinforcement learning environments.

1. **Permuted MNIST** (Goodfellow et al., 2013): a variant of the MNIST dataset (LeCun, 1998) where new tasks are created by permuting the pixels in all images in the same way. Each permutation represents a new task.

---

[3]Typical choices are $1/n$, 1, etc. (Dangel et al., 2025).

2. **Continual Imagenet** (Dohare et al., 2024): an adaptation of Imagenet (Krizhevsky et al., 2012) where pairs of classes are randomly sampled for binary classification. Distinguishing the two classes in each new class pair is a new task.

3. **Incremental CIFAR** (Dohare et al., 2024): an adaptation of the CIFAR-100 dataset (Krizhevsky et al., 2009) where classes are added incrementally. Starting with 5 classes, each task adds 5 new classes until all 100 classes are included.

4. **Slippery Ant** (Dohare et al., 2024): a continual reinforcement learning environment where the friction between the ant and the ground changes every $T$ timesteps, forcing the agent to continually adapt its policy.

We compare $L2$-ER against four baselines: vanilla backpropagation (BP), backpropagation with standard $L2$ regularization (L2), backpropagation with effective rank regularization (ER), and continual backpropagation (Dohare et al., 2024) (CBP). In Incremental CIFAR, tasks become more difficult as the number of classes to be classified increases. To decorrelate performance loss due to loss of plasticity from increasing task difficulty, we include an additional baseline (RESET), which reinitializes a new set of model parameters when task changes. Algorithms that perform better than RESET on Incremental CIFAR are considered to maintain plasticity.

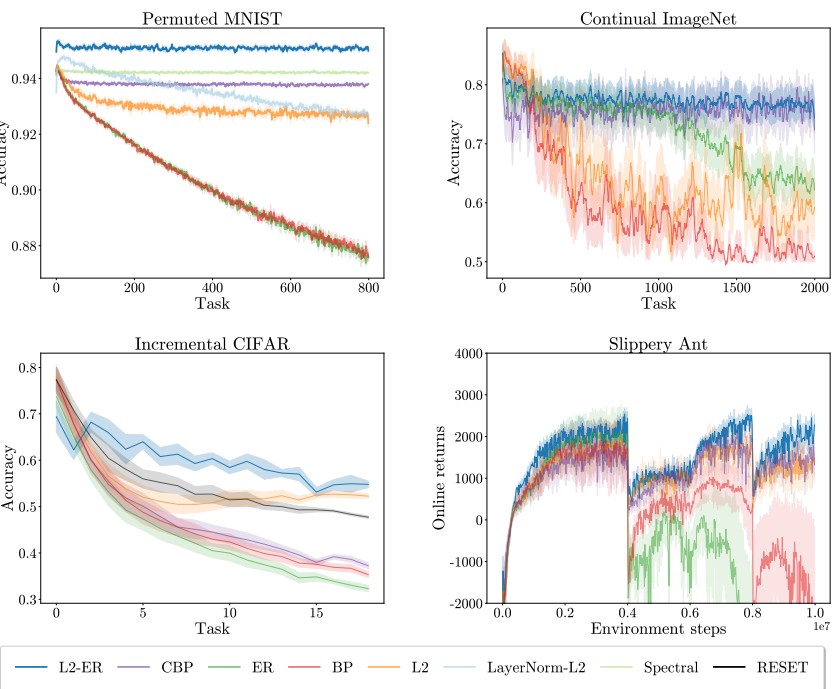

Figure 3: Performance across all four environments. Classification accuracy is reported for Permuted MNIST, Continual ImageNet, and Incremental CIFAR, while online returns are reported for Slippery Ant. Results are averaged across 5 seeds for all environments, except Continual ImageNet, where we used 10 seeds due to higher variance. Shaded regions denote a 95% confidence interval.

We performed an extensive hyperparameter sweep for each algorithm on all environments. We report accuracy for supervised learning environments and online returns for reinforcement learning environment. All experiments are written entirely in JAX (Bradbury et al., 2018), which allows for fast scalable experimentation; Full implementation details can be found in Appendices H to K.

**Performance** Figure 3 depicts our results across all environments. $L2$-ER shows strong performance, maintaining plasticity in all four environments. Standard $L2$ regularization also helps preserve plasticity in most cases, but fails on Continual ImageNet. CBP maintains plasticity on all environments except Incremental CIFAR. In contrast, ER seems to suffer from loss of plasticity in all environments, although it is able to maintain plasticity during the first half of training in

Continual ImageNet. Lastly, BP exhibits a severe loss of plasticity across all environments. On Permuted MNIST, $L2$-ER not only prevents loss of plasticity, but also provides a boost in performance at the start of training, well before loss of plasticity would affect outcomes.

**Single metrics are insufficient.** Consistent with Lyle et al. (2023), our results reinforce that no existing single pathology explains plasticity loss. In Figure 6, we plot effective feature rank and the number of dead neurons corresponding to the performance in Figure 3. While both metrics correlate with plasticity, neither is definitive. For example, ER maintains a high effective feature rank across environments yet still fails to preserve plasticity, whereas $L_2$ regularization exhibits low effective feature rank and many dead neurons but preserves plasticity in all environments except Continual ImageNet. Taken together, these findings indicate that the number of dead neurons and feature rank are, at best, partial proxies rather than reliable standalone indicators.

## 9 Conclusion

We identify Hessian spectral collapse as the central mechanism for plasticity loss in continual learning. We introduce $\tau$-trainability, proving that dead neurons persisting across tasks impose a probabilistic upper bound on the Hessian rank, thereby limiting the ability to learn. Using the KFAC approximation of the Hessian, we propose $L2$-ER, a simple regularizer that empirically prevents spectral collapse and preserves plasticity across diverse benchmarks. These results position spectral collapse as both a unifying explanation and a practical diagnostic for plasticity loss.

We leave adapting our theoretical framework of $\tau$-trainability to the unique dynamics of RL, where the Hessian is influenced by non-stationary policy and value functions for future work. Another critical direction is to investigate how spectral collapse manifests in attention-based models.

The implications of this work extend to the long-term vision for autonomous and adaptive AI systems (Silver & Sutton, 2025). Consider a large language model acting as a scientific assistant on a multi-year research project, or a robot deployed in a warehouse that must continually adapt to new products and workflows. The success of such long-horizon applications is fundamentally dependent on an agent's ability to integrate new information and refine its skills over its entire operational lifetime. By providing a stable foundation for lifelong learning, our findings represent a crucial step towards building the more capable and continuously adapting AI systems of the future.

## 10 REPRODUCIBILITY STATEMENT

We are committed to ensuring the reproducibility of our work. We provide an anonymized repository `https://anonymous.4open.science/r/lop-jax-E17B/README.md` with the full source code for our models, experiments, and evaluation. To support the reproduction of our empirical results, we detail all hyperparameter sweeps and selected configurations in the Appendices H to K. Complete proofs for all theoretical claims are also provided in the appendix.

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

APPENDIX

## A  Loss Landscape Visualizations

We follow the implementation of Li et al. (2018) to generate 3D loss landscapes for BP and $L2$-ER at task 800 initialization in Continual ImageNet. The resulting visualizations are shown in Figures 4 and 5. The BP landscape mostly have a 'taco-shaped' geometry, reflecting spectral collapse, whereas $L2$-ER yields a 'bowl-shaped' landscape with more curvature.

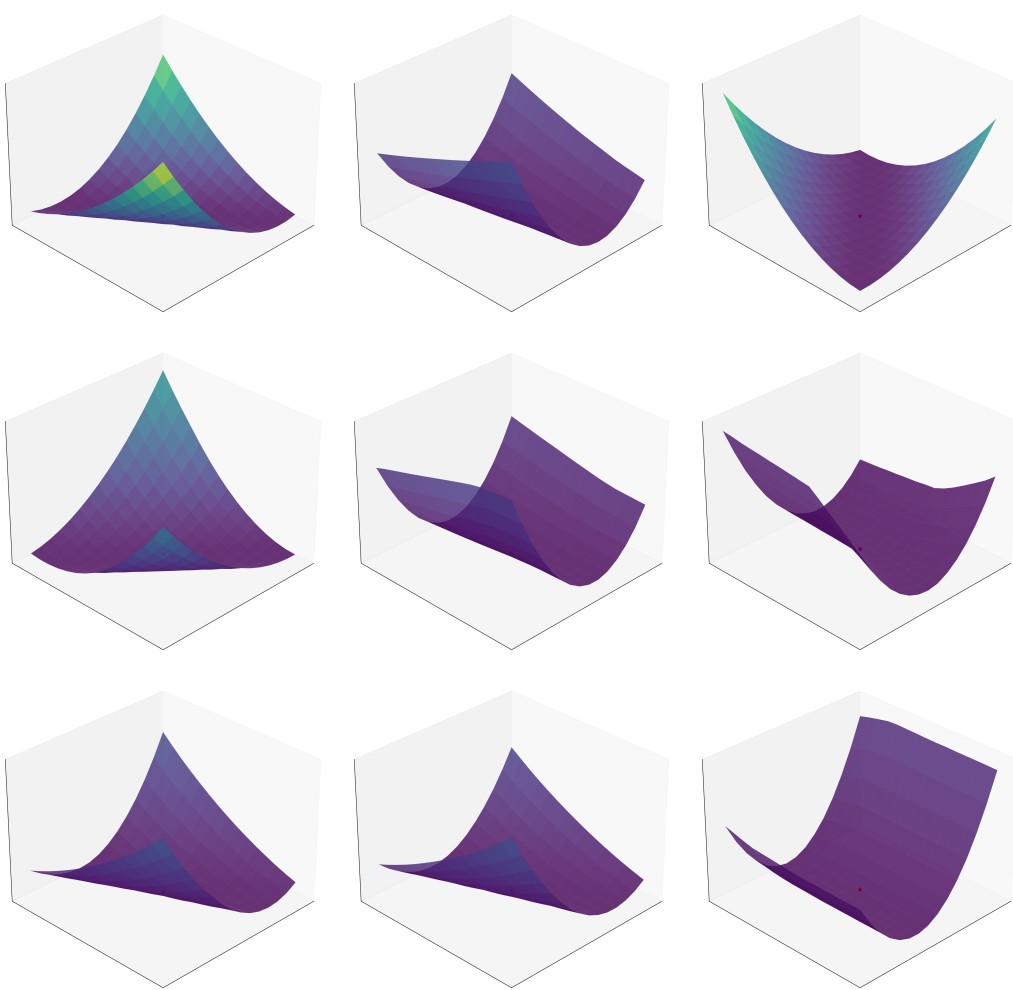

Figure 4: **BP:** 3D loss landscapes along the top principal directions at Continual Imagenet task 800 initialization

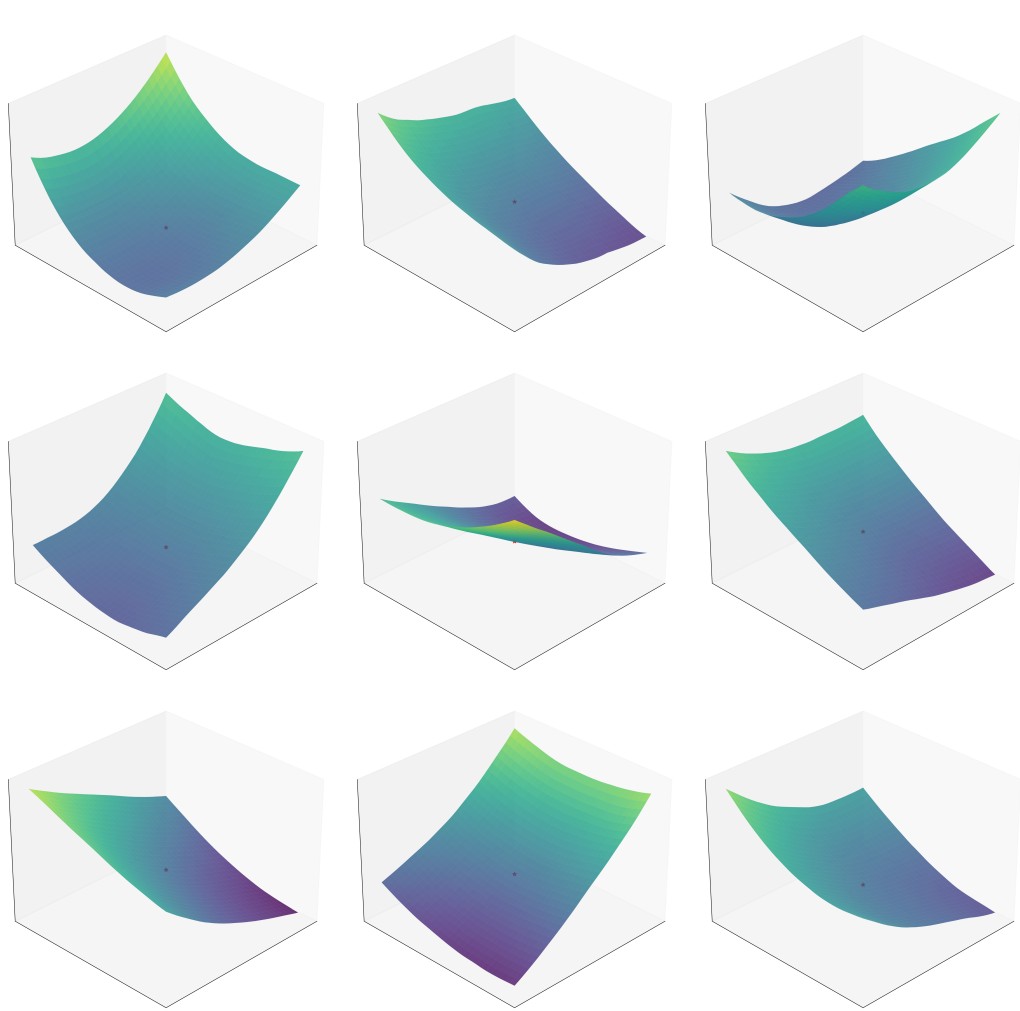

Figure 5: **L2-ER:** 3D loss landscapes are more bowl-shaped compared to BP, corresponding to higher curvature and therefore higher $\tau$-trainability.

# B    Proofs

## B.1    Proofs For Section 6

**Theorem 6.2.** *Let $P = H(I + 1) + O(H + 1)$ be the full parameter dimension, and let $k_\tau$ be the number of dead neurons on $\mathcal{X}_\tau$ at task $\tau$'s initialization. The Hessian rank satisfies $\text{rank}(\mathbf{H}_\tau^{(0)}) \leq P - k_\tau(I + O + 1)$. Equivalently, the number of dead neurons satisfies $k_\tau \leq \frac{P - \text{rank}(\mathbf{H}_\tau^{(0)})}{I + O + 1}$. Moreover,*

$$\mathbf{Pr}(T_\tau = 1) \leq \mathbf{Pr}\left(k_\tau \leq \frac{P - \rho_\tau}{I + O + 1}\right). \tag{1}$$

*Proof.* Let $J = \{j_1, \ldots, j_k\}$ index saturated units. For each $j \in J$, $\nabla_{W[[j,:],1]}\mathcal{L} = \mathbf{0}$, $\nabla_{\mathbf{b}_1[j]}\mathcal{L} = 0$, $\nabla_{\mathbf{w}_{[:,j],2}}\mathcal{L} = \mathbf{0}$. Since $\mathbf{W}_1 \in \mathbb{R}^H \times \mathbb{R}^I$, $\mathbf{W}_2 \in \mathbb{R}^O \times \mathbb{R}^H$, each dead neuron creates $D_j = I + O + 1$ 0-gradient directions. Therefore, with a little bit of re-arrangement, we can rewrite the neural network parameter as:

$$\theta = \left[\underbrace{\theta_{\text{active}}}_{P - kD_j}, \underbrace{\theta_{j_1}, \ldots, \theta_{j_k}}_{kD_j}\right]$$

where $\theta_j = [W_1[j,:], \mathbf{b}_1[j], W_2[:,j]]$ for $j \in J$. the Hessian $\nabla^2\mathcal{L}$ therefore exhibits a block structure

$$\nabla^2\mathcal{L} = \left(\begin{array}{c|c} \mathbf{A} & \mathbf{0} \\ \hline \mathbf{0} & \mathbf{0} \end{array}\right) \begin{array}{l} \} P - kD_j \\ \} kD_j \end{array}$$

Here $\mathbf{A} \in \mathbb{R}^{(P - kD_j) \times (P - kD_j)}$ and $\mathbf{0}$ blocks arise because:

- $\frac{\partial^2 \mathcal{L}}{\partial\theta_j \partial\theta_{\text{active}}} = \mathbf{0}$ (cross-terms vanish)

- $\frac{\partial^2 \mathcal{L}}{\partial\theta_j^2} = \mathbf{0}$ (flat directions)

Given a Hessian rank of $\text{rank}(\mathbf{H}_\tau)$ we can obtain the desired result since $\text{rank}(\mathbf{H}_\tau) \leq P - NULL(\nabla^2\mathcal{L}) \leq P - kD_j$. $\qquad\square$

**Lemma 6.3.** *Let $\mathcal{X}_\tau$ and $\mathcal{X}_{\tau'}$ be input datasets for two consecutive tasks. If neuron $j$ is $\gamma$-margin dead on $\mathcal{X}_\tau$, then neuron $j$ remains dead on $\mathcal{X}_{\tau'}$, if $\gamma_\tau^j \geq \Delta_\tau\|\mathbf{W}_{[j,:],1}\|$.*

*Proof.* By $\gamma_j$-margin dead definition:

$$\sup_{\boldsymbol{x} \in \mathcal{X}_\tau} h_j(\boldsymbol{x}) \leq -\gamma_j \tag{7}$$

where $h_j(\boldsymbol{x}) = \mathbf{W}_{[j,:]}\boldsymbol{x} + \mathbf{b}[j]$.

For any $\boldsymbol{x}' \in \mathcal{X}_{\tau'}$, Hausdorff distance guarantees:

$$\inf_{\boldsymbol{x} \in \mathcal{X}_\tau} \|\boldsymbol{x}' - \boldsymbol{x}\| \leq \Delta_\tau \tag{8}$$

Consider the pre-activation difference:

$$h_j(\boldsymbol{x}') - h_j(\boldsymbol{x}) = \mathbf{W}_{[j,:]}(\boldsymbol{x}' - \boldsymbol{x}) \tag{9}$$

By Cauchy-Schwarz and (8), $\exists\boldsymbol{x} \in \mathcal{X}_\tau$ such that:

$$\mathbf{W}_{[j,:]}(\boldsymbol{x}' - \boldsymbol{x}) \leq |\mathbf{W}_{[j,:]}(\boldsymbol{x}' - \boldsymbol{x})|$$
$$\leq \|\mathbf{W}_{[j,:]}\| \cdot \|\boldsymbol{x}' - \boldsymbol{x}\| \leq \|\mathbf{W}_{[j,:]}\|\Delta_\tau \tag{10}$$

Combining (9) and (10):

$$h_j(\boldsymbol{x}') \leq h_j(\boldsymbol{x}) + \|\mathbf{W}_{[j,:]}\|\Delta_\tau$$
$$\leq \sup_{\boldsymbol{x}\in\mathcal{X}_\tau} h_j(\boldsymbol{x}) + \|\mathbf{W}_{[j,:]}\|\Delta_\tau$$
$$\leq -\gamma_j + \|\mathbf{W}_{[j,:]}\|\Delta_\tau \quad \text{(by 7)} \tag{11}$$

Under the condition $\gamma_j \geq \Delta_\tau \|\mathbf{W}_{[j,:]}\|$:

$$h_j(\boldsymbol{x}') \leq -\gamma_j + \|\mathbf{W}_{[j,:]}\|\Delta_\tau \leq 0 \quad \forall \boldsymbol{x}' \in \mathcal{X}_{\tau'} \tag{12}$$

Thus neuron $j$ remains dead on $\mathcal{X}_{\tau'}$. $\qquad\square$

**Lemma B.1.** *Let each task-dataset consist of $N$ points drawn i.i.d. from the uniform distribution on the $I$-dimensional ball $\mathcal{B}_r(0)$. Write the Hausdorff distance between the two points clouds $\Delta_\tau := d_h(\mathcal{X}_\tau, \mathcal{X}_{\tau'})$. Then with probability at least $1 - \delta$,*

$$\Delta_\tau \leq r\left(\frac{\log(2N/\delta)}{N}\right)^{1/I} \tag{13}$$

*Furthermore, the remain dead probability $p_j$ satisfies*

$$p_j \geq 1 - 2\exp\left(\log N - N\left(\frac{\gamma_j}{\|\mathbf{W}_{[j,:]}\|r}\right)^I\right) \tag{14}$$

**Remark B.2.** *This lower bound approaches $1$ whenever*

$$N\left(\frac{\gamma_j}{\|\mathbf{W}_{[j,:]}\|\,r}\right)^I \gg \log N.$$

*In particular, it increases with larger margins $\gamma_j$, smaller weight norms $\|\mathbf{W}_{[j,:]}\|$, smaller domain radius $r$, lower dimension $I$, and larger sample size $N$. Intuitively, sufficiently negative pre-activations on task $\tau$, together with small inter-task drift, make the neuron stay inactive on the next task with high probability.*

**Proof:**

*Proof.* Fix $\epsilon \in (0, r]$. For a single point $x \in \mathcal{X}_\tau$, the probability that none of the N point from $\mathcal{X}_{\tau+1}$ falls inside $\mathcal{B}_\epsilon(x)$ is

$$\left(1 - \frac{V_d\epsilon^d}{V_d r^d}\right)^N = \left(1 - (\frac{\epsilon}{r})^d\right)^N \tag{15}$$

Taking a union bound over the N points in $\mathcal{X}_\tau$ and symmetrically over the N points in $\mathcal{X}_{\tau+1}$ gives

$$\mathbf{Pr}(\Delta_\tau > \epsilon) \leq 2N\left(1 - (\frac{\epsilon}{r})^d\right)^N \tag{16}$$
$$\leq 2N\left(\exp\left(-(\frac{\epsilon}{r})^d\right)\right)^N \text{ following } 1 + x \leq e^x \tag{17}$$
$$\leq 2\exp\left(\log N - N(\frac{\epsilon}{r})^d\right) \tag{18}$$

Then with probability at least $1 - \delta$,

$$\Delta_\tau \leq r\left(\frac{\log(2N/\delta)}{N}\right)^{1/d} \tag{19}$$

For the second claim, suppose neuron $j$ is $\gamma_j$-margin dead on $\mathcal{X}_\tau$, $|a_j(x') - a_j(x)| \leq \|\mathbf{W}_{[j,:]}\| \|x' - x\|$. Thus if $\Delta_\tau \leq \gamma_j/\|\mathbf{W}_{[j,:]}\|$, then every $x' \in \mathcal{X}_{\tau+1}$ has some $x \in \mathcal{X}_\tau$ with $a_j(x') \leq a_j(x) + \|\mathbf{W}_{[j,:]}\|\Delta_\tau \leq -\gamma_j + \gamma_j = 0$, so the neuron remains dead. Therefore

$$p_j = \mathbf{Pr}\left(\Delta_\tau \leq \frac{\gamma_j}{\|\mathbf{W}_{[j,:]}\|}\right) \geq 1 - 2\exp\left(\log N - N\left(\frac{\gamma_j}{\|\mathbf{W}_{[j,:]}\| r}\right)^I\right),$$

by substituting into the tail bound just proved

$\square$

**Theorem 6.5.** *Consider a 2-layer ReLU network with $n$ neurons and hidden layer dimension $H$ and a sequence of tasks $\tau \in \{1, \ldots, T\}$. At the end of task $\tau$, define $D_\tau$ to be the set of $\tau$-dead neurons (i.e., $|D_\tau| = k_\tau$). Let $m_{\tau'} = \frac{P - \rho_{\tau'}}{I + O + 1}$ be the maximum allowable number of dead neurons so that task $\tau'$ is trainable. If $\sum_j p_\tau^j \geq H - m_{\tau'}$, then the one-sided Chebyshev (Cantelli) inequality yields the following upper bound on $\tau'$-trainability:*

$$\mathbf{Pr}(T_{\tau'} = 1) \leq \mathbf{Pr}\left(\sum_j \xi_\tau^j \leq n - m_{\tau'}\right) \leq \frac{\mathbf{Var}(\sum_j \xi_\tau^j)}{\left[\sum_j p_\tau^j - (n - m_{\tau'})\right]^2}. \tag{2}$$

*Proof.* The total dead neurons at $\tau + 1$ initialization comprise:

$$k_{\tau+1} = k_d + \sum_{j \in D_\tau} \xi_j + \eta_{\tau+1} \tag{20}$$

where $\eta_{\tau+1}$ counts new dead neurons from previously active neurons. Since $\eta_{\tau+1} \geq 0$:

$$k_{\tau+1} \geq k_d + \sum_{j \in D_\tau} \xi_j \tag{21}$$

Taking expectations:

$$\mathbb{E}[k_{\tau+1}] \geq k_d + \sum_{j \in D_\tau} \mathbb{E}[\xi_j] = k_d + \sum_{j \in D_\tau} \mathbb{P}(\xi_j = 1) \tag{22}$$

Let $S_\tau = \sum_{j \in D_\tau} \xi_j$ with mean $\mu_S = \sum_j p_j$ and variance $\sigma_S^2 = \sum_j p_j(1 - p_j)$ where $p_j = \mathbb{P}(\xi_j = 1)$. Then

$$\mathbb{P}(\tau + 1 \text{ trainable}) \leq \mathbb{P}(S_\tau \leq n - m_{\tau+1} - k_d) \tag{23}$$

By Chebyshev's inequality:

$$\mathbb{P}(|S_\tau - \mu_S| \geq t) \leq \frac{\sigma_S^2}{t^2} \tag{24}$$

Setting $t = \mu_S - (n - m_{\tau+1} - k_d)$:

$$\mathbb{P}(S_\tau \leq n - m_{\tau+1} - k_d) \leq \frac{\sigma_S^2}{[\mu_s - (n - m_{\tau+1} - k_d)]^2} \tag{25}$$

$$= \frac{\sum_j p_j(1 - p_j)}{\left[\sum_j p_j - (n - m_{\tau+1} - k_d)\right]^2} \tag{26}$$

$\square$

B.2 PROOFS FOR SECTION 7

**Theorem B.3.** *Suppose an $L$-layered MLP $F_{\boldsymbol{\theta}} : \mathbb{R}^I \to \mathbb{R}^O$, and all input data lies in $\mathcal{B}_r(0)$, and either means-squared error or softmax-cross-entropy losses $\ell$. Let the network be $\boldsymbol{x} \mapsto F_{\boldsymbol{\theta}}(\boldsymbol{x})$ be $L_f$-Lipschitz on $\mathcal{B}_r(0)$ and let the logit-gradient, $\boldsymbol{z} \mapsto \nabla_{\boldsymbol{z}} \ell(\boldsymbol{z}, \boldsymbol{y})$ be $L_g$-Lipschitz. Define: $\Gamma(\boldsymbol{\theta}) := \sup_{\boldsymbol{x} \in \mathcal{B}_r(0), 1 \leq i \leq O} \|\nabla_{\boldsymbol{\theta}}^2 \boldsymbol{z}_i(\boldsymbol{x}; \boldsymbol{\theta})\|_2$ and $\boldsymbol{z} = F_{\boldsymbol{\theta}}(\boldsymbol{x})$ Then, the residual $\mathbf{R}$ of the Hessian on task $\tau + 1$ obeys the following operator-norm bound:*

$$\|\mathbf{R}(\boldsymbol{\theta}_{\tau+1}^0)\|_2 \leq \sqrt{O} \Gamma(\boldsymbol{\theta}_\tau) \left( \sup_{\boldsymbol{x} \in \mathrm{supp}(\mathcal{X}_\tau)} [\nabla_{\boldsymbol{z}} \ell(\boldsymbol{z}_i, \boldsymbol{y})] + L_f L_g d(\mathcal{X}, \mathcal{X}') \right) \doteq \boldsymbol{B}_\tau. \quad (27)$$

*Proof.* We start with the definition of the residual term of the Hessian:

$$\mathbf{R}(\boldsymbol{\theta}_{\tau+1}^0) = \mathbb{E}_{\boldsymbol{x} \sim \mathcal{X}_{\tau+1}} \left[ \sum_i^O \nabla_{\boldsymbol{z}} \ell(\boldsymbol{z}_i, \boldsymbol{y}) \nabla_{\boldsymbol{\theta}}^2 \boldsymbol{z}_i(\boldsymbol{x}; \boldsymbol{\theta}_\tau) \right] . \quad (28)$$

Next we apply linearity and operator norms:

$$\|\mathbf{R}(\boldsymbol{\theta}_{\tau+1}^0)\|_2 = \left\| \mathbb{E}_{\boldsymbol{x} \sim \mathcal{X}_{\tau+1}} \left[ \sum_i^O \nabla_{\boldsymbol{z}} \ell(\boldsymbol{z}_i, \boldsymbol{y}) \nabla_{\boldsymbol{\theta}}^2 \boldsymbol{z}_i(\boldsymbol{x}; \boldsymbol{\theta}_\tau) \right] \right\| \quad (29)$$

$$\leq \mathbb{E}_{\boldsymbol{x} \sim \mathcal{X}_{\tau+1}} \sum_i^O \|\nabla_{\boldsymbol{z}} \ell(\boldsymbol{z}_i, \boldsymbol{y})\|_2 \|\nabla_{\boldsymbol{\theta}}^2 \boldsymbol{z}_i(\boldsymbol{x}; \boldsymbol{\theta}_\tau)\|_2 \quad (30)$$

and apply a deterministic worst-case bound followed followed by the Cauchy-Schwartz inequality to obtain:

$$\|\mathbf{R}(\boldsymbol{\theta}_{\tau+1}^0)\|_2 \leq \sup_{\boldsymbol{x} \in \mathrm{supp}(\mathcal{X}')} \|\nabla_{\boldsymbol{z}} \ell(\boldsymbol{z}_i, \boldsymbol{y})\|_2 \sqrt{O} \Gamma(\boldsymbol{\theta}) \quad (31)$$

Let $\pi$ be a coupling measure on $\mathcal{X}$ and $\mathcal{X}'$, and denote $h(\boldsymbol{x}; \boldsymbol{\theta}) = \nabla_{\boldsymbol{z}} \ell(F_{\boldsymbol{\theta}}(\boldsymbol{x}), \boldsymbol{y})$. First operating inside the supp, we apply the triangle inequality:

$$\|\nabla_{\boldsymbol{z}} h(\boldsymbol{x}; \boldsymbol{\theta})\|_2 \leq \|h(\boldsymbol{x}'; \boldsymbol{\theta})\|_2 + \|h(\boldsymbol{x}; \boldsymbol{\theta}) - h(\boldsymbol{x}'; \boldsymbol{\theta})\|_2 \quad (32)$$

$$\leq \|h(\boldsymbol{x}'; \boldsymbol{\theta})\|_2 + L_f L_g d(\mathcal{X}, \mathcal{X}') \quad \text{Applying Lipschitz assumptions.} \quad (33)$$

Now, operating inside the supp w.r.t. $\pi$ we obtain:

$$\sup_{\boldsymbol{x} \in \mathrm{supp}(\mathcal{X})} \|h(\boldsymbol{x}; \boldsymbol{\theta})\|_2 = \sup_{(\boldsymbol{x}, \boldsymbol{x}') \in \mathrm{supp}(\pi)} \|h(\boldsymbol{x}'; \boldsymbol{\theta})\|_2 \leq \sup_{\boldsymbol{x}' \in \mathrm{supp}(\mathcal{X}')} \|h(\boldsymbol{x}'; \boldsymbol{\theta})\|_2 + L_f L_g d(\mathcal{X}, \mathcal{X}')$$

$$(34)$$

Finally, bounding the RHS of the inequality from above by an arbitrary constant $\boldsymbol{B}$ yields the desired result. $\qquad \square$

**Corollary B.4.** *We can use Weyl's inequality to show the existence of an appropriate $L2$ regularizer to guarantee that increasing $\mathrm{rank}(\mathbf{H}_{GGN})$ implies increasing the $\mathrm{rank}(\mathbf{H})$. Suppose ordered eigenvalues of $\mathbf{H}_{GGN}$ (i.e., $\lambda_1(\cdot) \geq \dots \lambda_n(\cdot)$), then by applying Wely's inequality we obtain*

$$|\lambda_i(\mathbf{H}_{GGN} + \mathbf{R}-)\lambda_i(\mathbf{R})| \leq \|\mathbf{R}\|_2 \quad (35)$$

$$\lambda_i(\mathbf{H}_{GGN} + \mathbf{R}) \geq \lambda_i(\mathbf{H}_{GGN}) - \|\mathbf{R}\|_2 \quad (36)$$

$$\lambda_i(\mathbf{H}) \geq \lambda_i(\mathbf{H}_{GGN}) - \|\mathbf{R}\|_2 \quad (37)$$

$$\lambda_{\min}^+(\mathbf{H}) \geq \lambda_{\min}^+(\mathbf{H}_{GGN}) - \|\mathbf{R}\|_2. \quad (38)$$

*The $L2$-regularized Hessian becomes:*

$$\lambda_{\min}^+(\mathbf{H} + 2\boldsymbol{I}q) \geq \lambda_{\min}^+(\mathbf{H}_{GGN}) - \|\mathbf{R}\|_2 + 2q \quad (39)$$

$$\geq \lambda_{\min}^+(\mathbf{H}_{GGN}) - \|\boldsymbol{B}\|_2 + 2q, \quad (40)$$

*so if $2q > \|\mathbf{R}\|_2 - \lambda_{\min}^+(\mathbf{H}_{GGN})$, then every positive eigenvalue of $\mathbf{H}_{GGN}$ remains positive in the $q$-regularized $\mathbf{H}$.*

**Remark B.5.** *Theorem B.3 and Corollary B.4 justify why $L2$ is needed alongside $ER$ regularization. Under softmax-cross-entropy loss and mean-squared-error loss with bounded targets, $h(\boldsymbol{x}; \boldsymbol{\theta})$ is also bounded which ensures the theorem is well-posed. The theorem also requires $\Gamma(\boldsymbol{\theta})$ to be uniformly bounded which can be achieved by techniques like spectral regularization (although we do use it in our experiments) (Yoshida & Miyato, 2017). In practice, we find that small choices of $L2$ regularization are needed which we report in table 2, table 4, and table 6.*

## C    Toy Landscapes, Task Transformation, and Curvature Regularization

Let $\theta = (\theta_1, \theta_2) \in \mathbb{R}^2$. Each task $\tau \in \{1, 2\}$ is built from two primitives: *canyon* $C_t$ (low curvature / low rank) and *bowl* $B_t$ (high curvature / high rank). The task loss is a soft minimum of the two:

$$L_t(\theta) = \text{soft} \min_{\tau_t}(C_t(\theta), B_t(\theta)) = -\tau_t \log\Big( \exp\big( - C_t(\theta)/\tau_t\big) + \exp\big( - B_t(\theta)/\tau_t\big)\Big), \quad (41)$$

where $\tau_t > 0$ controls the blend between components.

### C.1    Task 1 (base landscape)

$$
\begin{aligned}
C_1(\theta) &= c_{L1} \, (\theta_1 + 1)^2 \ + \ a_1 \, \theta_2^6, \\
B_1(\theta) &= c_{R1} \, (\theta_1 - x_{R1})^2 \ + \ b_1 \, (\theta_2 - y_{R1})^2.
\end{aligned}
\quad (42)
$$

We use $\tau_1 = 0.13$, $a_1 = 0.02$, $b_1 = 6.0$, $c_{L1} = 6.0$, $c_{R1} = 2.0$, $(x_{R1}, y_{R1}) = (0.8, 0.25)$. The low-curvature canyon minimum is near $(-1, 0)$ and the high-curvature bowl minimum near $(x_{R1}, y_{R1})$.

### C.2    Task 2 (explicit transformation of Task 1)

Task 2 preserves the structure in (41) but applies an affine rotation/translation to the canyon and relocates the bowl. Let $\mu = (cLx_2, cLy_2)$ and $R_\phi = \left( \begin{smallmatrix} \cos\phi & \sin\phi \\ -\sin\phi & \cos\phi \end{smallmatrix} \right)$. Define local canyon coordinates $\left( \begin{smallmatrix} u \\ v \end{smallmatrix} \right) = R_\phi(\theta - \mu)$. Then

$$
\begin{aligned}
C_2(\theta) &= c_u \, u^2 \ + \ a_2 \, v^6 \ + \ \varepsilon \, v^2, \\
B_2(\theta) &= b_x \, (\theta_1 - x_{R2})^2 \ + \ b_y \, (\theta_2 - y_{R2})^2.
\end{aligned}
\quad (43)
$$

We use $\tau_2 = 0.18$, $\phi = 35°$, $\mu = (-4.1, 2.0)$, $c_u = 10.0$, $a_2 = 10^{-4}$, $\varepsilon = 10^{-6}$, $(x_{R2}, y_{R2}) = (1.8, -0.6)$, and $(b_x, b_y) = (5.0, 9.0)$. Thus, the $L_1 \to L_2$ map is explicitly $\theta \mapsto (u, v) = R_\phi(\theta - \mu)$ in $C_2$, with a separate relocation of $B_2$.

### C.3    Curvature regularizer and the optimized objective

To promote high-curvature (well-conditioned) basins, we penalize flatness via a log-barrier on the Hessian eigenvalues of the task loss $L_t$:

$$R_{\text{curve}}(\theta; L_t) := - \sum_{i=1}^{2} \log(\lambda_i(\theta; L_t)^2 + \epsilon), \quad (44)$$

where $\lambda_i(\theta; L_t)$ are the eigenvalues of $\nabla^2 L_t(\theta)$ and $\epsilon > 0$ ensures stability (we use $\epsilon = 10^{-6}$). The optimized objective for *curvature-regularized GD* is

$$F_t(\theta) \ = \ L_t(\theta) \ + \ \alpha_t \, R_{\text{curve}}(\theta; L_t) \ + \ \frac{\beta}{2} \, \|\theta\|_2^2, \quad (45)$$

with small isotropic shrinkage $\beta > 0$ (we use $\beta = 0.006$) and task-dependent weight $\alpha_t$ (we use $\alpha_1 = 0.12$, $\alpha_2 = 0.10$).

**Gradient update.**    Both baselines use identical step sizes $\eta$; the only difference is the objective ($L_t$ vs. $F_t$):

$$\text{GD:} \quad \theta_{k+1} = \theta_k - \eta \, \nabla L_t(\theta_k), \quad (46)$$

$$\text{Curv-Reg-GD:} \quad \theta_{k+1} = \theta_k - \eta \Big( \nabla L_t(\theta_k) + \alpha_t \, \nabla R_{\text{curve}}(\theta_k; L_t) + \beta \, \theta_k \Big). \quad (47)$$

In our toy implementation, $\nabla L_t$ and $\nabla R_{\text{curve}}$ are computed via finite differences of the corresponding scalars.

# D    KFAC ERROR

For a given layer, $\mathbf{W}_l \in \mathbb{R}^{d_{in} \times d_{out}}$, with activations into the layer $\mathbf{a}_l \in \mathbb{R}^{d_{in}}$ and gradient outputs $\boldsymbol{g} \in \mathbb{R}^{d_{out}}$. The respective covariances are $\mathbf{a}_l \mathbf{a}_l^\top \in \mathbb{R}^{d_{in} \times d_{in}}$ and $\boldsymbol{g}_l \boldsymbol{g}_l^\top \in \mathbb{R}^{d_{out} \times d_{out}}$. The resulting KFAC block is of size $(d_{out} d_{in}) \times (d_{out} d_{in})$. The per-entry error, on the $n$th sample, can be obtained by indexing over Kronecker matrix pairs with row-index $(i, \alpha)$ and column index $(j, \beta)$. We use $\cdot_{l,[i,*]}$ to denote the $i$th row of $(\cdot)$ in layer $l$, and $\cdot_{l,[*,j]}$ for the $j$th column. Flattening over the Kronecker product, we obtain the following for a given entry in layer $l$:

$$(\mathbf{H}_{GGN})_{l,[i,\alpha],[j,\beta]} = \frac{1}{N} \sum_{n=1}^{N} \sum_{o=1}^{O} \mathbf{a}_{l[i,*]}^n \mathbf{a}_{l[*,j]}^n \boldsymbol{g}_{l[\alpha,*]}^{n,o} \boldsymbol{g}_{l[*,\beta]}^{n,o} = \overline{\mathbf{a}_{l[i,*]} \mathbf{a}_{l[*,j]} \boldsymbol{g}_{l[\alpha,*]} \boldsymbol{g}_{l[*,\beta]}} \tag{48}$$

$$(\widehat{\mathbf{H}}_{GGN})_{l,[i,\alpha],[j,\beta]} = \left( \frac{1}{N} \sum_{m=1}^{N} \mathbf{a}_{l[i,*]}^n \mathbf{a}_{l[*,j]}^n \right) \cdot \left( \frac{1}{N} \sum_{n=1}^{N} \sum_{o=1}^{O} \boldsymbol{g}_{l[\alpha,*]}^{m,o} \boldsymbol{g}_{l[*,\beta]}^{m,o} \right) = \overline{\mathbf{a}_{l[i,*]} \mathbf{a}_{l[*,j]}} \cdot \overline{\boldsymbol{g}_{l[\alpha,*]} \boldsymbol{g}_{l[*,\beta]}}, \tag{49}$$

where we use $\overline{\cdot}$ to denote the empirical sample average. In this way, we can compute the error as the difference between the average of products and the product of averages; for a fixed index quadruple $(i, j, \alpha, \beta)$, the KFAC error of the $((i, \alpha), (j, \beta))$th-entry is:

$$\Delta_{[i,\alpha],[j,\beta]} = (\mathbf{H}_{GGN} - \widehat{\mathbf{H}}_{GGN})_{[i,\alpha],[j,\beta]} = \overline{\mathbf{a}_{l[i,*]} \mathbf{a}_{l[*,j]} \boldsymbol{g}_{l[\alpha,*]} \boldsymbol{g}_{l[*,\beta]}} - \overline{\mathbf{a}_{l[i,*]} \mathbf{a}_{l[*,j]}} \cdot \overline{\boldsymbol{g}_{l[\alpha,*]} \boldsymbol{g}_{l[*,\beta]}} \tag{50}$$

# E ALGORITHM

## E.1 BASELINES

**Backpropagation (BP):** We optimize the objective with stochastic gradient descent (SGD) on cross-entropy loss over the current task. Let $\theta$ denote all trainable parameters and $\mathcal{D}_t$ the data for task $t$. The objective is

$$\min_\theta \ \mathbb{E}_{(x,y)\sim\mathcal{D}_t}[L(f_\theta(x), y)].$$

**Vanilla Effective Rank (ER):** In addition to BP, we collect the output features of each dense layer (excluding convolutional layers) over a fixed number of steps (er-batch in Table 1). From these stacked features, we compute the effective rank (Roy & Vetterli, 2007) for each layer and sum across all layers. Then we maximize the effective rank of the network representations.

$$\mathcal{L}_{\mathrm{ER}} = -\frac{1}{L}\sum_{\ell=1}^{L}\mathrm{ER}(F_\ell), \qquad \mathrm{ER}(F_\ell) = \exp\Big(-\sum_{i=1}^{d_\ell} p_i^{(\ell)}\log p_i^{(\ell)}\Big),$$

where $F_\ell \in \mathbb{R}^{n\times d_\ell}$ is the stacked feature matrix for layer $\ell$, $s_i(F_\ell)$ are its singular values, and

$$p_i^{(\ell)} = \frac{s_i(F_\ell)}{\sum_j s_j(F_\ell)}.$$

**L2 normalization (L2):** We add weight decay to BP. Now the objective becomes:

$$\min_\theta \ \mathbb{E}_{(x,y)\sim\mathcal{D}_t}[L(f_\theta(x), y)] \ + \ \lambda_w\|\theta\|_2^2,$$

where $\lambda_w$ is the weight-decay coefficient (weight-decay in Table 1).

**Continual Backpropagation (CBP):** We follow the architecture in Dohare et al. (2024).

## E.2 EFFECTIVE RANK WITH $L2$ NORMALIZATION ($L2$-ER)

The pseudocode for our implementation of $L2$-ER is shown in Algorithm 1. $L2$-ER augments the standard task objective with two additional regularizers: (1) an effective rank penalty and (2) an $L2$ weight decay term. At each step, the task loss $L_{\mathrm{task}}$ together with the weight decay term is minimized via standard backpropagation. Simultaneously, the hidden features $F_\ell$ are collected over a fixed window L of updates. Every L steps, these stacked features are used to compute the effective rank (Roy & Vetterli, 2007) of the representation at each layer. The effective rank losses are averaged across layers. Note that a gradient descent step is taken on this loss only every L steps.

---

**Algorithm 1** Continual Learning with Effective-Rank Regularization ($L2$-ER)

---

**Input:** Task datasets $\{\{(\boldsymbol{x}_\tau^i, \boldsymbol{y}_\tau^i)\}_{i=0}^{N-1}\}_{\tau=0}^{T-1}$; model $f(\cdot; W)$; learning rates $\alpha$ (task), $\beta$ (ER); weight decay $\lambda$; ER update interval $U$ (in steps)
**State:** Per-layer feature buffers $\{\mathcal{B}_\ell\}_{\ell=1}^{L}$ with capacity $U$
**for** $\tau = 0$ **to** $T-1$ **do**
    **for** $i = 0, \ldots, N-1$ **do**
        $(\hat{\boldsymbol{y}}, \{F_\ell\}_{\ell=1}^{L}) \leftarrow f(\boldsymbol{x}_\tau^i; W)$ {$\hat{\boldsymbol{y}}$ is the model prediction}
        $\forall\ell: \ \mathcal{B}_\ell \leftarrow \mathrm{enqueue}(\mathcal{B}_\ell, F_\ell)$; if $|\mathcal{B}_\ell| > U$ then drop oldest
        $L_{\mathrm{task}} \leftarrow \mathrm{Loss}(\hat{\boldsymbol{y}}, \boldsymbol{y}_\tau^i) + \lambda\|W\|_2^2$
        $W \leftarrow W - \alpha\nabla_W L_{\mathrm{task}}$
        **if** $(i+1) \bmod U = 0$ **then**
            $L_{\mathrm{erank}} \leftarrow -\frac{1}{L}\sum_{\ell=1}^{L}\mathrm{ER}(\mathrm{SVD}(\mathrm{Stack}(\mathcal{B}_\ell)))$ {(Roy & Vetterli, 2007)}
            $W \leftarrow W - \beta\nabla_W L_{\mathrm{erank}}$
            $\mathcal{B}_\ell \leftarrow \emptyset$
        **end if**
    **end for**
**end for**
**Return:** $W$

---

# F   DEAD NEURONS AND EFFECTIVE RANK

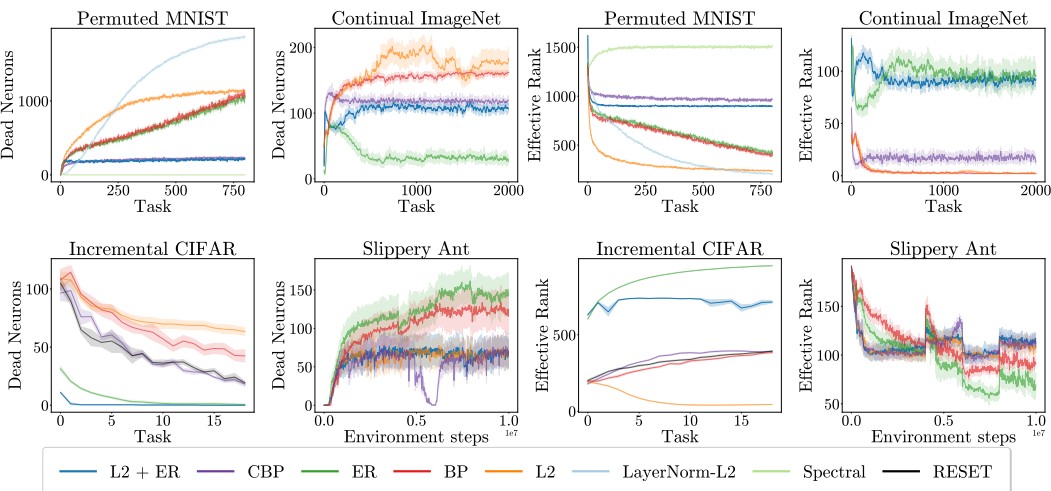

Figure 6: Number of dead neurons (left) and effective rank (right) corresponding to Figure 3.

Here, we present number of dead neurons and effective rank corresponding to Figure 3. An important observation to note here is that dead neurons in Incremental Cifar is decreasing rather than increasing. This effect arises due to the following two reasons: First, at the beginning of training, the environment contains only 5 classes, which gradually increase to 100. As more classes are introduced, the evaluation set becomes much larger, increasing the likelihood of encountering samples that activate a given neuron (i.e., produce nonzero outputs). Second, to accommodate the changing number of classes, we mask the outputs of the final layer to match the number of active classes in each task. In the early tasks, this masking leads to a sharp rise in the number of dead neurons, since the network tends to overfit to the small set of available classes. As more classes are added, this effect diminishes. Therefore, we also measured the number of dead neurons and effective rank in RESET for comparison. Learning curves that lie above RESET indicate an increase in dead neurons due to loss of plasticity, whereas curves below suggest relatively fewer dead neurons.

## G   Epsilon Hessian Rank

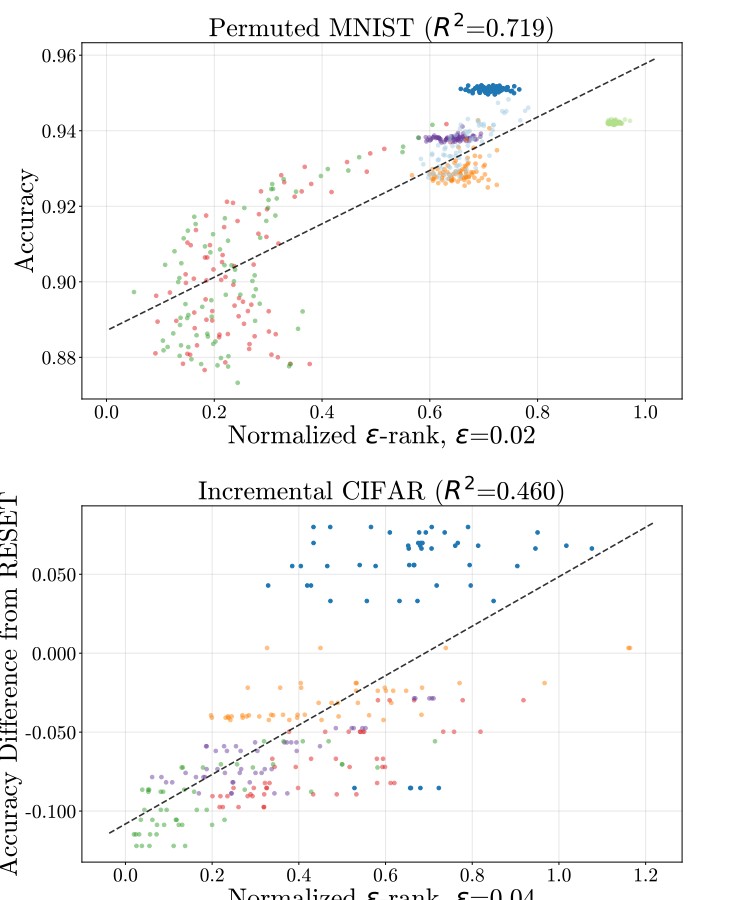

Figure 7: Curvature vs. Accuracy on Permuted MNIST (Left) and First 10 Task Incremental CIFAR (Right). A linear fit (dotted line) highlights the positive association between curvature and accuracy.

### G.1   Measuring the Hessian Spectral Density

To empirically demonstrate the relationship between dead ReLUs and the rank of the Hessian, we estimate the Hessian spectrum over continual learning tasks (Ghorbani et al., 2019). The spectral density is defined as $\psi(u) = \frac{1}{P} \sum_{i=1}^{P} \delta(u - \lambda_i)$ where $\delta$ is a Dirac delta operator. Since the naive approach to estimating the density would involve calculating every eigenvalue, we turn to a Gaussian relaxation (Ghorbani et al., 2019):

$$\psi_\sigma(u) = \frac{1}{P} \sum_{i=1}^{P} f(\lambda_i; u, \sigma^2)$$

where $f(\lambda_i; u, \sigma^2) = \frac{1}{\sigma\sqrt{2\pi}} \exp(-\frac{(u-\lambda)^2}{2\sigma^2})$ and $\sigma^2$ is the variance. When $\sigma^2$ is small, $\psi_\sigma$ provides a tight estimate of the spectral density. Since materializing the full Hessian is prohibitively expensive, we estimate the density with the stochastic Lanczos quadrature algorithm (Golub & Welsch, 1969; Ghorbani et al., 2019), which takes advantage of the fact that accessing Hessian-vector-products (HVP) is a computationally efficient operation (Pearlmutter, 1994). Given $\mathbf{H}$ is diagonalized and $f$ has a closed-form equation, we can define $f(\mathbf{H}) = \mathbf{Q} f(\mathbf{\Lambda}) \mathbf{Q}^\top$ where $f(\mathbf{\Lambda})$ acts on the diagonal.

Now we estimate the spectrum of Hessian through the HVP with $\mathbf{v} \sim N(0, \frac{1}{P}\boldsymbol{I}_{P \times P})$ which gives:

$$\psi_\sigma = \frac{1}{P}\text{tr}\left(f(\mathbf{H}, u, \sigma^2)\right) = \mathbb{E}[\mathbf{v}^\top f(\mathbf{H}, u, \sigma^2)\mathbf{v}]$$

In practice, each $\psi_\sigma^{\mathbf{v}}$ is approximated by $m$-steps of the Lanczos algorithm resulting in a $m \times m$ tridiagonal matrix with $m$ locations-weight pairs $(\ell_j, \omega_j)$ so that:

$$\psi_\sigma^{\mathbf{v}}(u) \approx \sum_{j=1}^{m} \omega_j f(\ell_j; u, \sigma^2) \doteq \hat{\psi}^{\mathbf{v}}(u)$$

Moreover, $\psi_\sigma^{\mathbf{v}}(\cdot)$ is an unbiased estimator of $\psi_\sigma(\cdot)$ and $\hat{\psi}^{\mathbf{v}}(u)$ converges exponentially fast around its expectation over samples of $\mathbf{v}$ (Ghorbani et al., 2019)[Claim 2.3] resulting in a computationally efficient and accurate estimation of the spectral density, even for large neural networks.

Throughout our experiments, we measure the Hessian eigen-spectrum at the beginning and at the end of each new task. Our results show that the spectrum of standard back-propagation narrows as the task number grows. In fact, a complete loss of learning corresponds to a complete collapse of the spectrum. Furthermore, we show that loss of plasticity mitigation like continual backpropagation and our own $L2$-ER method preserve the spectrum.

In the following sections, we present the details of each environment, their corresponding hyperparameter sweeps and selected best hyperparameters, followed by the analysis of the Hessian spectrum.

### G.2 EPSILON HESSIAN RANK VS ACCURACY

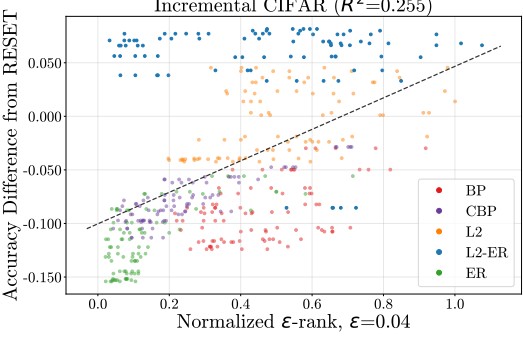

Figure 8: Curvature vs. Accuracy on Full 20 Task Incremental CIFAR. A linear fit (dotted line) highlights the positive association between curvature and accuracy.

In addition to Figure 2, we provide the epsilon rank of the other two supervised learning environments in Figure 7. Continual ImageNet and Permuted MNIST's data points on the graph are calculated by an average across 5 seeds.

Putting all the Incremental Cifar tasks into one plot does not yield as clear a positive relationship as Permuted MNIST or Continual Imagenet. This is due to the offset in measuring both successful training and $\tau$-trainability. All the tasks in Permuted MNIST or Continual Imagenet are about the same level of difficulties for $\tau \in \{1, \cdots, T\}$, which gives us the ability to easily measure successful training through task accuracies. In Incremental Cifar, each task $\tau$ is composed of classifying $5 * \tau$ classes of images using the same computational budget, which makes it a challenge to isolate unsuccessful training due to loss of plasticity from increasing task difficulties. We report the performance difference between algorithms and a freshly initialized network as a measure of successful training. Furthermore, neural network tends to find a lower-rank solution regardless of initialization Hu et al. (2022). Since we always report the $\epsilon$-Hessian rank of the Full databatch, the percentage of data the neural network has already been trained on becomes larger and larger as $\tau$

grows. For example, at task 2 initialization, we trained on 5 classes and acquired a low rank representation, then 5 classes are added and the $\epsilon$-Hessian rank is high; while at task 20 initialization, we trained on 19 classes and are evaluating on 20, this is basically the same as the eigenspectrum at task 19 convergence, which is a low rank solution. In short, due to the non-uniform property of the tasks, $\epsilon$-Hessian rank is not an accurate measure of Spectral Collapse in Incremental Cifar. When we group the first ten tasks (fig. 7), we can see that the positive relationship is clearer since the task difficulties are more similar.

# H    PERMUTED MNIST

We now detail environments and architectural details in all experiments. All algorithms are written in JAX (Bradbury et al., 2018). Although each environment is independent, our training procedure is designed to be easily generalizable. In the following sections, we provide detailed descriptions of each environment.

Permuted MNIST (Dohare et al., 2024) is a variant of the original MNIST dataset (LeCun, 1998) where the pixels are permuted randomly in the same way for each image in the original dataset. Each permutation defines a new task for the learner. In total, we evaluate on 800 tasks, each of which is a 10-class classification problem.

## H.1    NETWORK ARCHITECTURE

We employ a standard multilayer perceptron (MLP) with three hidden layers of hidden size 1000 each followed by a ReLU activation. All weights are initialized with Kaiming uniform. The architecture can be summarized as follows:

```
MLP(
  Sequential(
    (0): Dense(init=kaiming_uniform, out_dims=1000, bias=True)
    (1): ReLU()
    (2): Dense(init=kaiming_uniform, out_dims=1000, bias=True)
    (3): ReLU()
    (4): Dense(init=kaiming_uniform, out_dims=1000, bias=True)
    (5): ReLU()
    (6): Dense(init=kaiming_uniform, out_dims=10, bias=True)
  )
)
```

## H.2    HYPERPARAMETERS

In Table 1, we present the default hyperparam of our experiments. Unless otherwise specified, experiments use the default hyperparameters.

## H.3    EXPERIMENTS

Prior work (Dohare et al., 2024), along with our own experiments, shows that the learning rate is a critical factor in continual learning: using a smaller learning rate consistently yields only marginal differences in performance. To better highlight the phenomenon of loss of plasticity, we fix the learning rate to $1 \times 10^{-2}$ and sweep over the remaining hyperparameters in Table 2. We report the results in Figure 6 and the selected best hyperparameters in Table 2. For completeness of our experiments, we also evaluate the best learning rate for each algorithm by sweeping all the hyperparameters in Table 2, with results shown in Figure 9 and the corresponding hyperparameters summarized in Table 2.

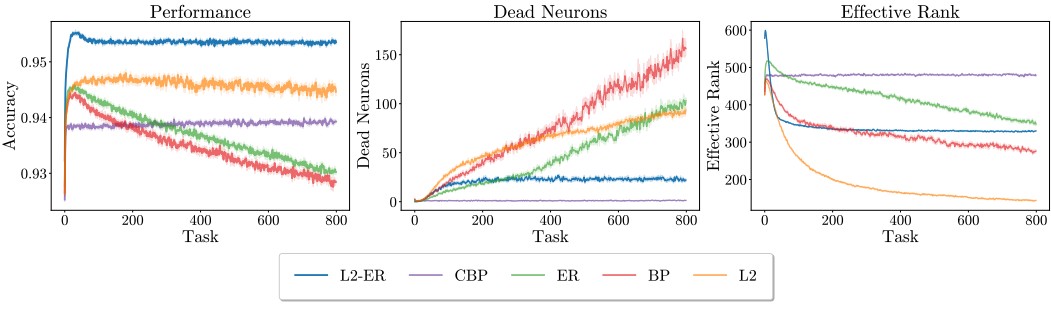

Figure 9: Best Learning Rate Performance, dead neurons, and effective rank in Permuted MNIST.

| Hyperparam Name | Default | Description |
|---|---|---|
| study_name | test | experiment name |
| seed | 2024 | base random seed |
| debug | False | true to enable debug mode |
| platform | gpu | {cpu, gpu} |
| n_seeds | 1 | number of seeds |
| env | permuted_mnist | environment name |
| agent | l2_er | agent options: {er, bp, l2, snp_l2, snp, cbp, l2_er} |
| activation | relu | activation options: {relu, tanh} |
| lr | [0.01] | learning rate(s) |
| optimizer | sgd | {adam, sgd} |
| weight_decay | 0.001 | $L2$ weight decay |
| num_features | 1000 | hidden size for the mlp |
| change_after | $10 \times 6000$ | steps between task switches |
| to_perturb | False | whether to perturb the input data |
| perturb_scale | $1 \times 10^{-5}$ | magnitude of input perturbation |
| num_hidden_layers | 3 | number of hidden layers in the mlp |
| mini_batch_size | 1 | minibatch size |
| no_anneal_lr | True | if true, do not anneal the learning rate |
| max_grad_norm | 0.5 | gradient clipping threshold |
| num_tasks | 800 | number of tasks used in training/eval |
| **effective rank** | | |
| er_lr | [0.01] | ER learning rate |
| er_batch | 100 | batch size for er computation |
| er_step | 1 | ER update frequency |
| **evaluation** | | |
| evaluate | True | evaluate after each task |
| evaluate_previous | False | evaluate on previous task |
| eval_size | 2000 | number of evaluate samples per task |
| compute_hessian | False | whether to compute hessian spectrum |
| compute_hessian_size | 2000 | samples used for hessian computation |
| compute_hessian_interval | 1 | interval in tasks between hessian runs |
| **continual backpropagation** | | |
| cont_backprop | False | enable CBP |
| replacement_rate | $1 \times 10^{-6}$ | CBP replacement probability per step |
| decay_rate | 0.99 | exponential decay for CBP statistics |
| maturity_threshold | 100 | steps before a unit is considered "mature" |
| **Spectral Regularizer** | | |
| k | 2 | power used in $(\sigma_{\max}^k - \texttt{target})^2$ |
| target | 2.0 | target value for the largest singular value |
| spectral_strength | 0.1 | coefficient multiplying the spectral penalty |
| num_iter | 10 | number of power-iteration steps to estimate $\sigma_{\max}$ |

Table 1: permuted MNIST default hyperparameters

### H.4 PERMUTED MNIST HESSIAN SPECTRUM

We use the best hyperparameters in Table 2 and rerun it with 5 seeds to plot the hessian spectrum during our training. We calculate the approximated hessian matrix every fixed number of tasks (compute-hessian-interval in Table 1). The corresponding performance is shown in Figure 3 and Hessian spectrum of seed 2025 is shown in Figure 10, where the orange curve corresponds to the Hessian spectrum on the test set and the blue curve corresponds to the training set. Since plotting all tasks is impractical, we present only a subset of representative tasks.

Comparing Figure 3 with Figure 10, we observe that algorithms which eventually lose plasticity exhibit a sparse Hessian spectrum. In contrast, algorithms that maintain plasticity preserve a rich and dense spectrum, proving our claim that spectral collapse is strongly correlated with the loss

| Algorithm | Hyperparameter | Sweep Hyperparameters | Fixed LR Best | Best |
|---|---|---|---|---|
| BP | Learning rate | $\{1 \times 10^{-2}, 1 \times 10^{-3}, 1 \times 10^{-4}\}$ | $1 \times 10^{-2}$ | $1 \times 10^{-3}$ |
| ER | Learning rate | $\{1 \times 10^{-2}, 1 \times 10^{-3}, 1 \times 10^{-4}\}$ | $1 \times 10^{-2}$ | $1 \times 10^{-3}$ |
| | Effective rank lr | $\{1 \times 10^{-2}, 1 \times 10^{-3}, 1 \times 10^{-4}\}$ | $1 \times 10^{-3}$ | $1 \times 10^{-3}$ |
| $L2$-ER | Learning rate | $\{1 \times 10^{-3}\}$ | $1 \times 10^{-2}$ | $1 \times 10^{-3}$ |
| | Effective rank lr | $\{1 \times 10^{-2}, 1 \times 10^{-3}, 1 \times 10^{-4}\}$ | $1 \times 10^{-3}$ | $1 \times 10^{-3}$ |
| | Weight decay | $\{1 \times 10^{-3}, 1 \times 10^{-4}, 1 \times 10^{-5}\}$ | $1 \times 10^{-3}$ | $1 \times 10^{-3}$ |
| CBP | Learning rate | $\{1 \times 10^{-2}, 1 \times 10^{-3}, 1 \times 10^{-4}\}$ | $1 \times 10^{-2}$ | $1 \times 10^{-3}$ |
| | Replacement rate | $\{1 \times 10^{-4}, 1 \times 10^{-5}, 1 \times 10^{-6}\}$ | $1 \times 10^{-6}$ | $1 \times 10^{-6}$ |
| $L2$ | Learning rate | $\{1 \times 10^{-2}, 1 \times 10^{-3}, 1 \times 10^{-4}\}$ | $1 \times 10^{-2}$ | $1 \times 10^{-3}$ |
| | Weight decay | $\{1 \times 10^{-3}, 1 \times 10^{-4}, 1 \times 10^{-5}\}$ | $1 \times 10^{-3}$ | $1 \times 10^{-3}$ |
| LayerNorm-L2 | Learning rate | $\{1 \times 10^{-2}, 1 \times 10^{-3}, 1 \times 10^{-4}\}$ | $1 \times 10^{-2}$ | |
| | Weight decay | $\{1 \times 10^{-2}, 1 \times 10^{-3}, 1 \times 10^{-4}\}$ | $1 \times 10^{-5}$ | |
| Spectral Regularizer | Learning rate | $\{1 \times 10^{-2}, 1 \times 10^{-3}, 1 \times 10^{-4}\}$ | $1 \times 10^{-2}$ | |
| | Spectral Strength | $\{1 \times 10^{-2}, 1 \times 10^{-3}, 1 \times 10^{-4}\}$ | $1 \times 10^{-3}$ | |

Table 2: Hyperparameter sweeps and best values for Permuted MNIST across all algorithms, with and without fixing the learning rate.

of plasticity. Furthermore, even subtle reductions in plasticity are reflected in the spectrum: for instance, $L2$ shows a slight decline of about 1% in accuracy over training. This small change is captured by the eigenspectrum, as shown in Figure 10, where the range of eigenvalues for $L2$ is narrower than that of the other two algorithms that preserve plasticity.

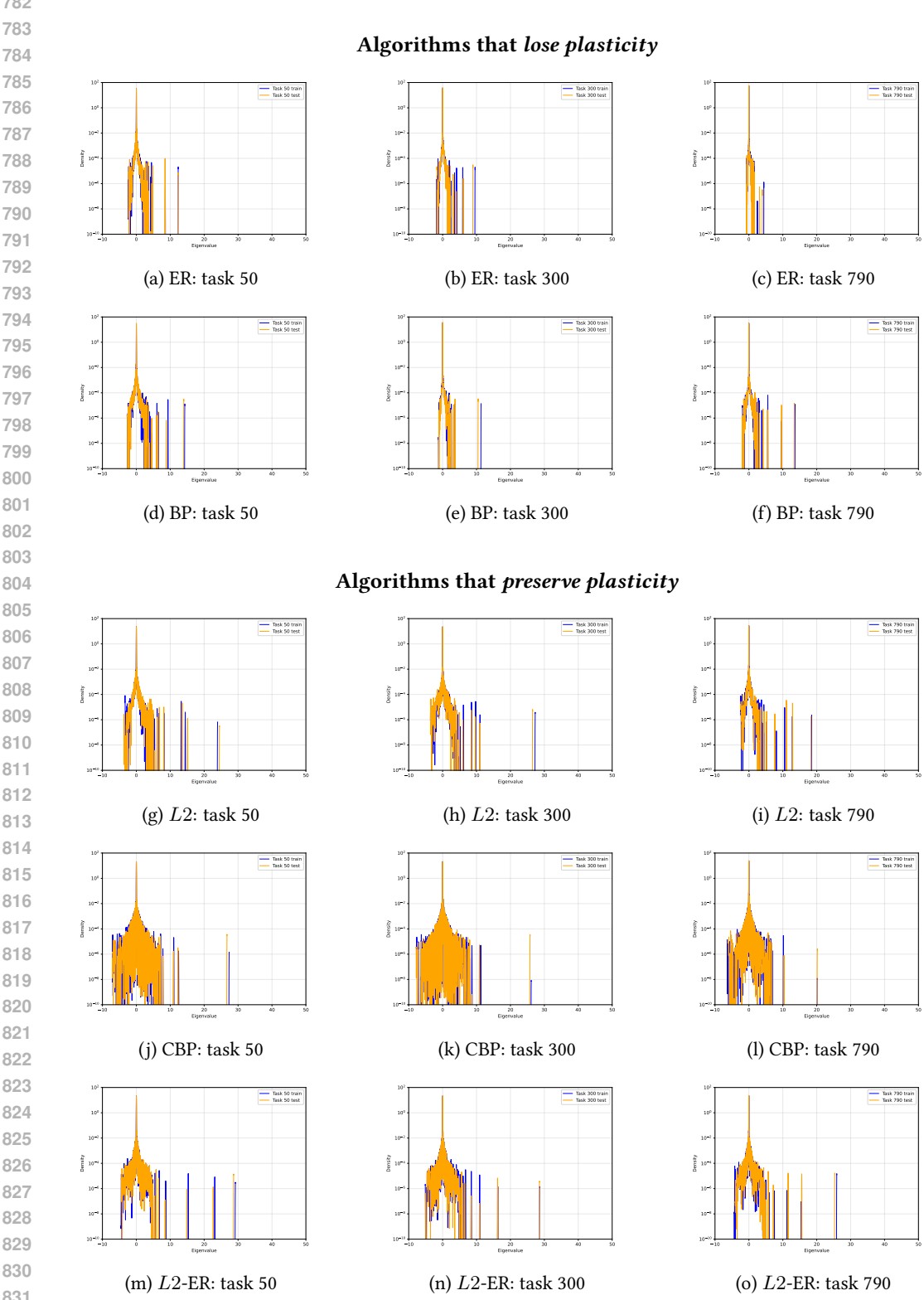

Figure 10: Comparison of Hessian eigenspectra at **task init** across permuted MNIST. From top to bottom, the algorithms are ordered by increasing accuracy. **Top:** Algorithms that lose plasticity (ER, BP). **Bottom:** Algorithms that preserve plasticity ($L2$, CBP, $L2$-ER).

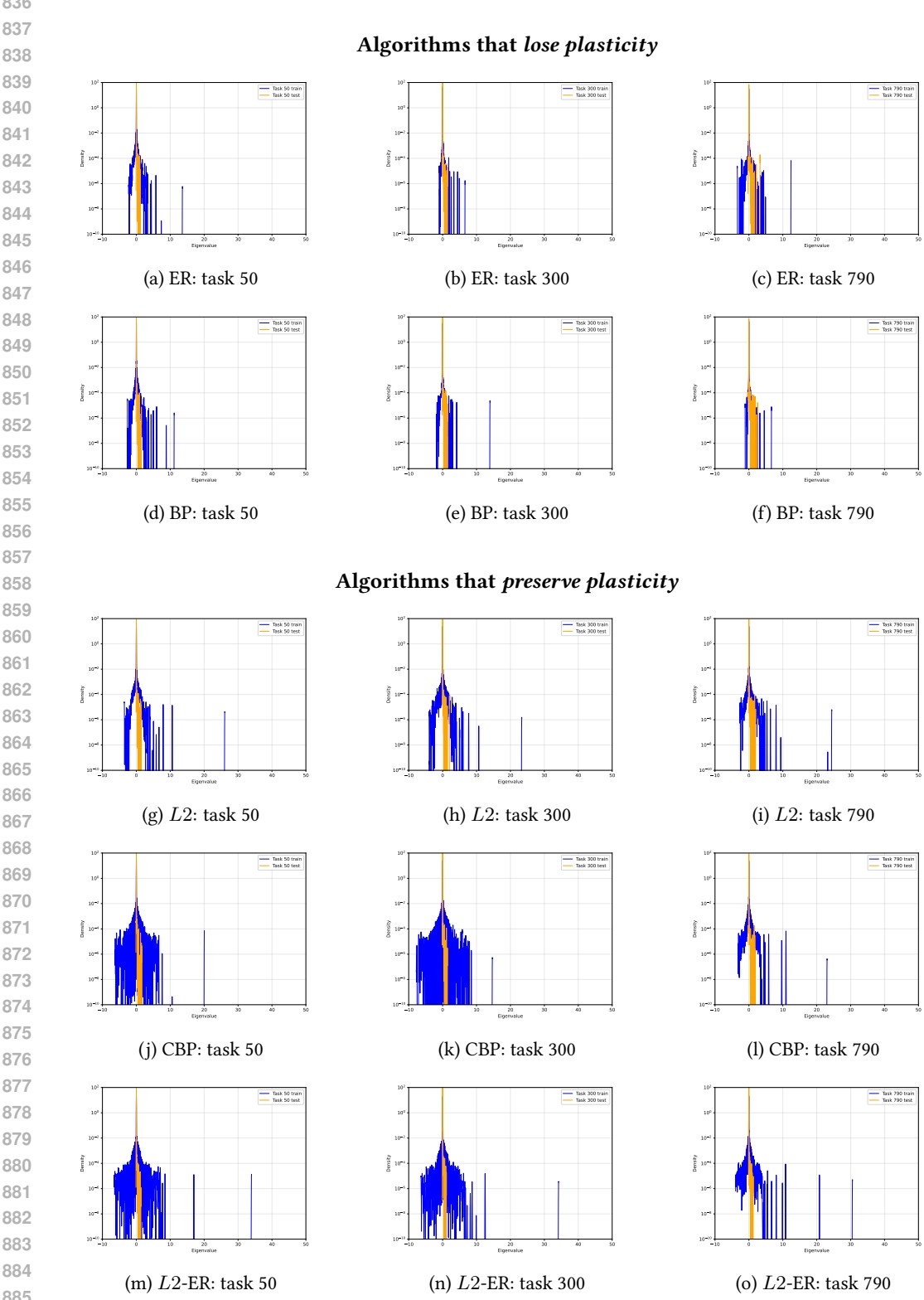

Figure 11: Comparison of Hessian eigenspectra at **task end** across permuted MNIST. From top to bottom, the algorithms are ordered by increasing accuracy. **Top:** Algorithms that lose plasticity (ER, BP). **Bottom:** Algorithms that preserve plasticity ($L2$, CBP, $L2$-ER).

## I  CONTINUAL IMAGENET

Continual ImageNet (Dohare et al., 2024) is an adaptation of ImageNet (Krizhevsky et al., 2012) in which pairs of classes are randomly sampled to form binary classification tasks. We evaluate performance on a total of 2000 tasks.

### I.1  NETWORK ARCHITECTURE

We adopt the same architecture as Dohare et al. (2024) with one key modification. In their setup, the final layer of the network is reinitialized at the beginning of every task. To ensure fairness in comparison, we remove this feature and keep the final layer fixed across tasks. We use a three–block convolutional network followed by two dense layers and a dense classifier. Each convolution is followed by a ReLU nonlinearity and a $2 \times 2$ max-pool with stride 2. The architecture is summarized as follows:

```
ConvNet(
  Sequential(
    (0): Conv2d(out_channels=32, kernel_size=5x5, bias=True)
    (1): ReLU()
    (2): MaxPool(window=2x2, stride=2)

    (3): Conv2d(out_channels=64, kernel_size=3x3, bias=True)
    (4): ReLU()
    (5): MaxPool(window=2x2, stride=2)

    (6): Conv2d(out_channels=128, kernel_size=3x3, bias=True)
    (7): ReLU()
    (8): MaxPool(window=2x2, stride=2)
    (9): Flatten()

    (10): Dense(out_dims=128, bias=True)
    (11): ReLU()
    (12): Dense(out_dims=128, bias=True)
    (13): ReLU()
    (14): Dense(out_dims=2, bias=True)
  )
)
```

### I.2  HYPERPARAMETERS

In Table 3, we present the default hyperparameters for our ImageNet experiments. Unless otherwise specified, experiments use these defaults.

### I.3  EXPERIMENTS

Due to high variance, we conducted full hyperparameter sweeps from Table 4 across 10 seeds and apply Savitzky-Golay filter (Savitzky & Golay, 1964) to the results. The best hyperparameters selected from these sweeps are summarized in Table 4.

### I.4  CONTINUAL IMAGENET HESSIAN SPECTRUM

We again use the best hyperparameters from Table 4 to run over 10 seeds to plot the hessian spectrum. We presents our results in Figure 13 and its corresponding performance in Figure 3. Note that we categorize ER as preserving plasticity in this case because, in this single run, ER successfully maintained plasticity rather than losing it. We present this single run hessian spectrum accuracy in Figure 12. From Figure 13, we observe that all algorithms that lose plasticity experience spectral collapse, whereas those that preserve plasticity maintain a wide and dense spectrum throughout training.

| Hyperparam Name | Default | Description |
|---|---|---|
| study_name | test | experiment name |
| seed | 2024 | base random seed |
| debug | False | true to enable debug mode |
| platform | gpu | {cpu, gpu} |
| n_seeds | 1 | number of seeds |
| env | imagenet | environment name |
| agent | l2_er | agent options: {er, bp, l2, snp_l2, snp, cbp, l2_er} |
| alg | ppo | algorithm type: {actor_critic, ppo} |
| activation | relu | activation options: {relu, tanh} |
| lr | [0.01] | learning rate(s) |
| optimizer | sgd | {adam, sgd} |
| weight_decay | 0.001 | $L2$ weight decay |
| mini_batch_size | 100 | minibatch size |
| no_anneal_lr | True | if true, do not anneal the learning rate |
| max_grad_norm | $1 \times 10^9$ | gradient clipping threshold |
| num_tasks | 2000 | number of tasks used in training/eval |
| num_epochs | 250 | number of epochs per task |
| momentum | 0.9 | SGD momentum coefficient |
| **effective rank** | | |
| er_lr | [0.01] | ER learning rate |
| er_batch | 1 | batch size for ER computation |
| er_step | 1 | ER update frequency |
| **evaluation** | | |
| evaluate | True | evaluate after each task |
| eval_size | 2000 | number of evaluation samples per task |
| compute_hessian | False | whether to compute hessian spectrum |
| compute_hessian_size | 2000 | samples used for hessian computation |
| compute_hessian_interval | 1 | interval in tasks between hessian runs |
| **continual backpropagation** | | |
| cont_backprop | False | enable CBP |
| replacement_rate | $1 \times 10^{-6}$ | CBP replacement probability per step |
| decay_rate | 0.99 | exponential decay for CBP statistics |
| maturity_threshold | 100 | steps before a unit is considered "mature" |

Table 3: ImageNet default hyperparameters

| Algorithm | Hyperparameter | Sweep Hyperparameters | Best Hyperparam |
|---|---|---|---|
| BP | Learning rate | $10^{-2}$, $10^{-3}$, $10^{-4}$ | $10^{-4}$ |
| ER | Learning rate | $10^{-2}$, $10^{-3}$, $10^{-4}$ | $10^{-4}$ |
| | Effective rank lr | $10^{-3}$, $10^{-4}$, $10^{-5}$ | $10^{-5}$ |
| $L2$-ER | Learning rate | $10^{-3}$ | $10^{-3}$ |
| | Effective rank lr | $10^{-3}$, $10^{-4}$, $10^{-5}$ | $10^{-4}$ |
| | Weight decay | $10^{-3}$, $10^{-4}$, $10^{-5}$ | $10^{-3}$ |
| CBP | Learning rate | $10^{-2}$, $10^{-3}$, $10^{-4}$ | $10^{-4}$ |
| | Replacement rate | $10^{-4}$, $10^{-5}$, $10^{-6}$ | $10^{-5}$ |
| $L2$ | Learning rate | $10^{-2}$, $10^{-3}$, $10^{-4}$ | $10^{-4}$ |
| | Weight decay | $10^{-3}$, $10^{-4}$, $10^{-5}$ | $10^{-3}$ |

Table 4: Hyperparameter sweeps and selected best values for Continual ImageNet across all algorithms.

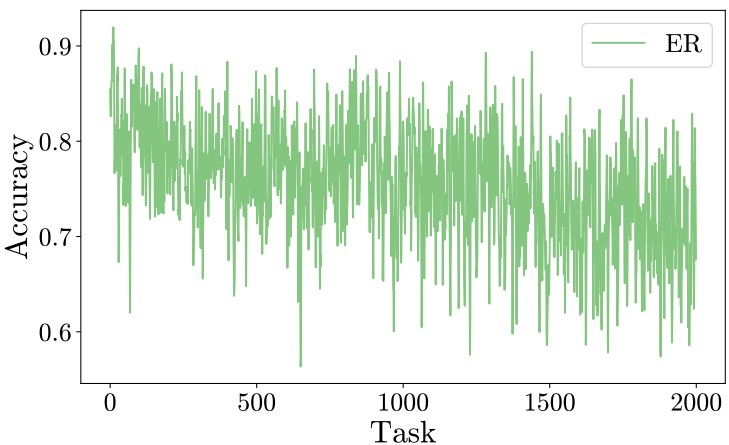

Figure 12: ER Accuracy correspond to Hessian Spectrum on Continual ImageNet.

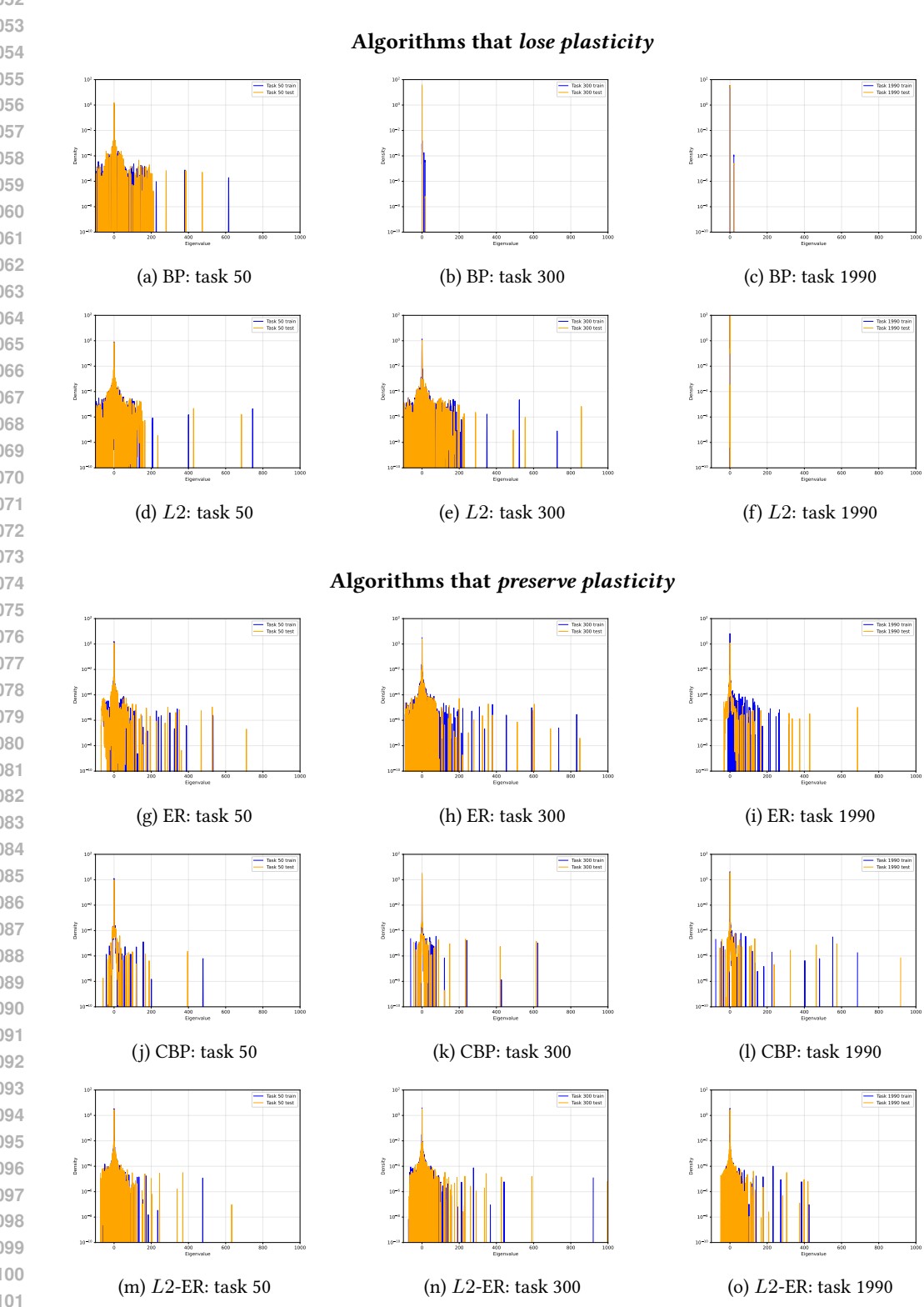

Figure 13: Comparison of Hessian eigenspectra at task init on Continual ImageNet. From top to bottom, the algorithms are ordered by increasing accuracy. **Top:** Algorithms that lose plasticity (ER, BP). **Bottom:** Algorithms that preserve plasticity ($L2$, CBP, $L2$-ER).

## J  INCREMENTAL CIFAR

Incremental CIFAR (Dohare et al., 2024) is an adaptation of the original CIFAR-100 dataset (Krizhevsky et al., 2009) where classes are incrementally added. Starting with 5 classes, each task adds 5 new classes for classification until all 100 classes are included. Our setup largely follows Dohare et al. (2024), but we remove random data transformations and learning rate annealing from their design.

### J.1  NETWORK ARCHITECTURE

We employ a standard ResNet-18 architecture adapted from Dohare et al. (2024). The network begins with a $3 \times 3$ convolutional stem with 64 channels, followed by four sequential residual stages. Each stage contains two BasicBlocks: Layer1 keeps the width at 64 channels, while Layers 2–4 progressively double the number of channels to 128, 256, and 512, with the first block of each stage performing downsampling via stride-2 convolutions and $1 \times 1$ skip connections. After the residual stages, a global average pooling layer reduces the spatial dimension, and then pass the resulting feature vector through two fully connected layers with ReLU activations. Finally, a dense classification head outputs logits for the number of classes in the current task.

```
ResNet18(
  Stem(
    Conv2d(out_channels=64, kernel_size=3x3, stride=1, padding=1, bias=True)
    BatchNorm()
    ReLU()
  )
  Layer1: Sequential(
    BasicBlock(64 -> 64, stride=1)(
      Conv2d(64, 3x3, stride=1, padding=1, bias=True), BatchNorm(), ReLU,
      Conv2d(64, 3x3, stride=1, padding=1, bias=True), BatchNorm(),
      Skip: Identity,
      Add, ReLU
    )
    BasicBlock(64 -> 64, stride=1)(
      Conv2d(64, 3x3, stride=1, padding=1, bias=True), BatchNorm(), ReLU,
      Conv2d(64, 3x3, stride=1, padding=1, bias=True), BatchNorm(),
      Skip: Identity,
      Add, ReLU
    )
  )
  Layer2: Sequential(
    BasicBlock(64 -> 128, stride=2)(
      Conv2d(128, 3x3, stride=2, padding=1, bias=True), BatchNorm(), ReLU,
      Conv2d(128, 3x3, stride=1, padding=1, bias=True), BatchNorm(),
      Skip: Conv2d(128, 1x1, stride=2, bias=True) + BatchNorm(),
      Add, ReLU
    )
    BasicBlock(128 -> 128, stride=1)(
      Conv2d(128, 3x3, stride=1, padding=1, bias=True), BatchNorm(), ReLU,
      Conv2d(128, 3x3, stride=1, padding=1, bias=True), BatchNorm(),
      Skip: Identity,
      Add, ReLU
    )
  )
  Layer3: Sequential(
    BasicBlock(128 -> 256, stride=2)(
      Conv2d(256, 3x3, stride=2, padding=1, bias=True), BatchNorm(), ReLU,
      Conv2d(256, 3x3, stride=1, padding=1, bias=True), BatchNorm(),
      Skip: Conv2d(256, 1x1, stride=2, bias=True) + BatchNorm(),
      Add, ReLU
```

```
      )
      BasicBlock(256 -> 256, stride=1)(
        Conv2d(256, 3x3, stride=1, padding=1, bias=True), BatchNorm(), ReLU,
        Conv2d(256, 3x3, stride=1, padding=1, bias=True), BatchNorm(),
        Skip: Identity,
        Add, ReLU
      )
    )
    Layer4: Sequential(
      BasicBlock(256 -> 512, stride=2)(
        Conv2d(512, 3x3, stride=2, padding=1, bias=True), BatchNorm(), ReLU,
        Conv2d(512, 3x3, stride=1, padding=1, bias=True), BatchNorm(),
        Skip: Conv2d(512, 1x1, stride=2, bias=True) + BatchNorm(),
        Add, ReLU
      )
      BasicBlock(512 -> 512, stride=1)(
        Conv2d(512, 3x3, stride=1, padding=1, bias=True), BatchNorm(), ReLU,
        Conv2d(512, 3x3, stride=1, padding=1, bias=True), BatchNorm(),
        Skip: Identity,
        Add, ReLU
      )
    )
    GlobalAveragePool()
    Dense(out_dims=512, bias=True), ReLU()
    Dense(out_dims=512, bias=True), ReLU()
    Dense(out_dims=num_classes, bias=True)
)
```

## J.2    Hyperparameters

In Table 5, we present the default hyperparameters for our ImageNet experiments. Unless otherwise specified, experiments use these defaults.

## J.3    Experiments

We conduct our hyperparameter sweep experiments over 5 seeds for all hyperparameters listed in Table 6, and report the selected best hyperparameters in Table 6.

## J.4    Incremental CIFAR hessian spectrum

We use the best hyperparameters in Table 6 to plot the Hessian spectrum over 5 seeds, with results shown in Figure 14. Again consistent with our previous experiments, algorithms that preserve plasticity maintain a dense Hessian spectrum, whereas those that lose plasticity exhibit a sparse spectrum. This demonstrates our claim that spectral collapse is a clear indicator of plasticity loss.

| Hyperparam Name | Default | Description |
|---|---|---|
| study_name | test | experiment name |
| seed | 2024 | base random seed |
| debug | False | true to enable debug mode |
| platform | gpu | {cpu, gpu} |
| n_seeds | 1 | number of seeds |
| env | incremental_cifar | environment name |
| agent | l2 | agent options: {er, bp, l2, snp_l2, snp, cbp, l2_er} |
| alg | ppo | algorithm type: {actor_critic, ppo} |
| activation | relu | activation options: {relu, tanh} |
| lr | [0.1] | learning rate(s) |
| optimizer | sgd | {adam, sgd} |
| weight_decay | 0.0005 | $L2$ weight decay |
| to_perturb | False | whether to perturb the input data |
| perturb_scale | $1 \times 10^{-5}$ | magnitude of input perturbation |
| mini_batch_size | 100 | minibatch size |
| no_anneal_lr | False | if true, do not anneal the learning rate |
| max_grad_norm | $1 \times 10^{9}$ | gradient clipping threshold |
| num_tasks | 20 | number of tasks used in training/eval |
| num_epochs | 200 | number of epochs per task |
| momentum | 0.9 | SGD momentum coefficient |
| **effective rank** | | |
| er_lr | [0.01] | ER learning rate |
| er_batch | 5 | batch size for ER computation |
| er_step | 1 | ER update frequency |
| **resetting** | | |
| reset | False | reset the network after each task |
| **evaluation** | | |
| evaluate | True | evaluate after each task |
| evaluate_previous | False | evaluate on previous tasks |
| eval_size | 2000 | number of evaluation samples per task |
| compute_hessian | False | whether to compute Hessian spectrum |
| compute_hessian_size | 2000 | samples used for Hessian computation |
| compute_hessian_interval | 1 | interval in tasks between Hessian runs |
| **continual backpropagation** | | |
| cont_backprop | False | enable CBP |
| replacement_rate | $1 \times 10^{-6}$ | CBP replacement probability per step |
| decay_rate | 0.99 | exponential decay for CBP statistics |
| maturity_threshold | 100 | steps before a unit is considered "mature" |

Table 5: Incremental CIFAR default hyperparameters

| Algorithm | Hyperparameter | Sweep Values | Best Value |
|---|---|---|---|
| BP | Learning rate | $\{1 \times 10^{-2}, 1 \times 10^{-3}, 1 \times 10^{-4}\}$ | $1 \times 10^{-2}$ |
| ER | Learning rate | $\{1 \times 10^{-2}, 1 \times 10^{-3}, 1 \times 10^{-4}\}$ | $1 \times 10^{-2}$ |
| | Effective rank lr | $\{1 \times 10^{-3}, 1 \times 10^{-4}, 1 \times 10^{-5}\}$ | $1 \times 10^{-4}$ |
| $L2$-ER | Learning rate | $\{1 \times 10^{-2}\}$ | $1 \times 10^{-2}$ |
| | Effective rank lr | $\{1 \times 10^{-2}, 1 \times 10^{-3}, 1 \times 10^{-4}\}$ | $1 \times 10^{-3}$ |
| | Weight decay | $\{1 \times 10^{-3}, 1 \times 10^{-4}, 1 \times 10^{-5}\}$ | $1 \times 10^{-3}$ |
| CBP | Learning rate | $\{1 \times 10^{-2}, 1 \times 10^{-3}, 1 \times 10^{-4}\}$ | $1 \times 10^{-2}$ |
| | Replacement rate | $\{1 \times 10^{-4}, 1 \times 10^{-5}, 1 \times 10^{-6}\}$ | $1 \times 10^{-6}$ |
| $L2$ | Learning rate | $\{1 \times 10^{-2}, 1 \times 10^{-3}, 1 \times 10^{-4}\}$ | $1 \times 10^{-2}$ |
| | Weight decay | $\{1 \times 10^{-3}, 1 \times 10^{-4}, 1 \times 10^{-5}\}$ | $1 \times 10^{-4}$ |

Table 6: Hyperparameter sweeps and selected best values for Incremental CIFAR across all algorithms.

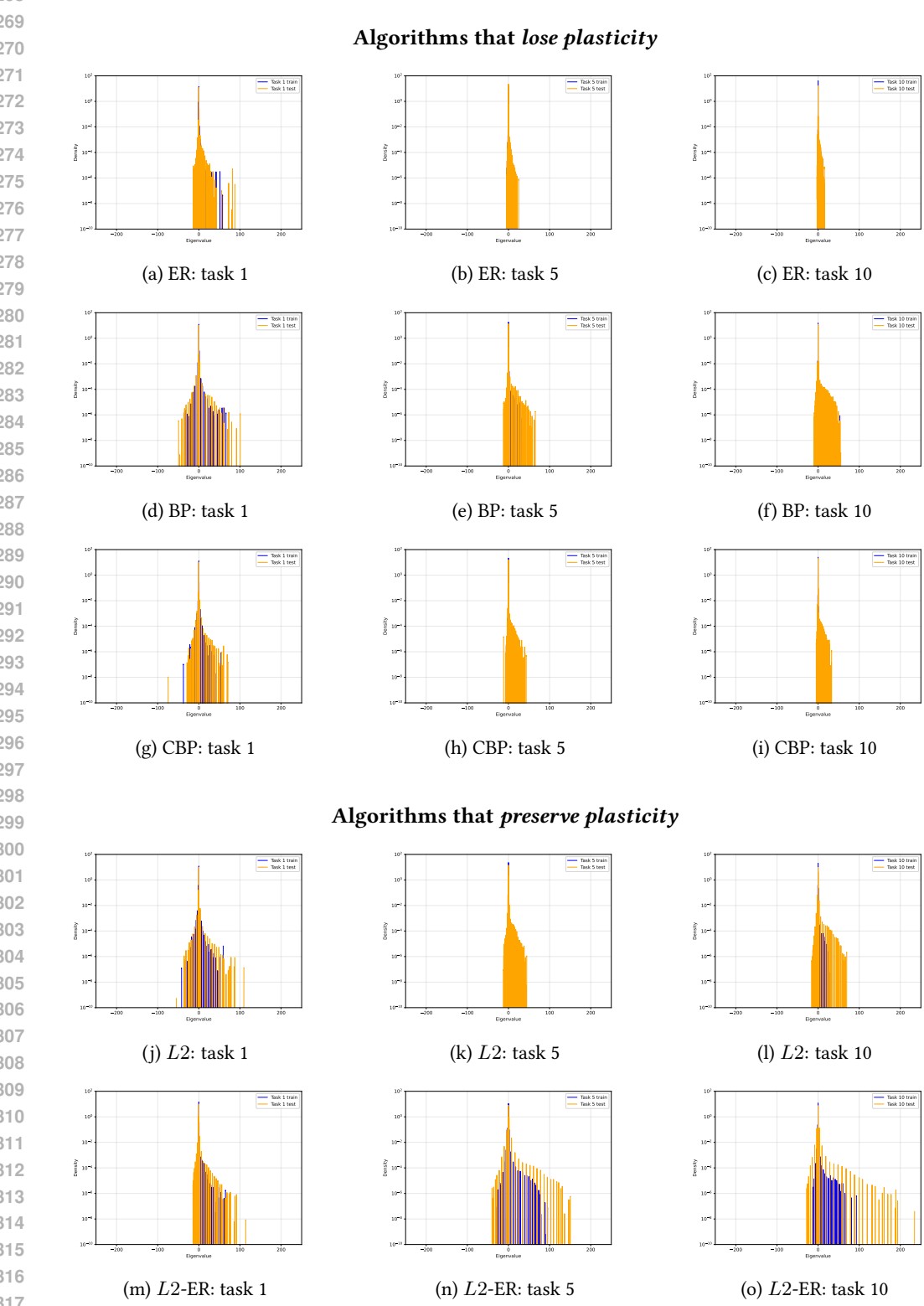

Figure 14: Comparison of Hessian eigenspectra on Incremental CIFAR. Algorithms are ordered by increasing accuracy. **Top:** Algorithms that lose plasticity (ER, BP, CBP). **Bottom:** Algorithms that preserve plasticity ($L2$, $L2$-ER).

## K  SLIPPERY ANT

Slippery Ant is a continual reinforcement learning environment where the friction between the ant and the ground changes every 2 million steps. The agent has to adapt its policy to this new friction. In our experiments, we use PPO (Schulman et al., 2017) as our base algorithm, which is an online learning method designed for training on vectorized environments. It is parallelized using the JAX library (Bradbury et al., 2018) based on a batch experimentation library written in JAX (Lu et al., 2022).

### K.1  NETWORK ARCHITECTURE

```
ActorCritic(
  Actor(
    Sequential(
      (0): Dense(in_dims=hidden_size, out_dims=hidden_size, bias=True)
      (1): ReLU()
      (2): Dense(in_dims=hidden_size, out_dims=action_dims, bias=True)
      (3): Categorical() or MultivariateNormalDiag()
    )
  )
  Critic(
    Sequential(
      (0): Dense(in_dims=hidden_size, out_dims=hidden_size, bias=True)
      (1): ReLU()
      (2): Dense(in_dims=hidden_size, out_dims=1, bias=True)
    )
  )
)
```

### K.2  HYPERPARAMETERS

The default hyperparameters is in Table 8. Unless otherwise specified, experiments use the default hyperparameters.

| Algorithm | Hyperparameter | Sweep Values | Best Value |
|---|---|---|---|
| BP | Learning rate | $\{1 \times 10^{-2}, 1 \times 10^{-3}, 1 \times 10^{-4}\}$ | $1 \times 10^{-4}$ |
| ER | Learning rate | $\{1 \times 10^{-2}, 1 \times 10^{-2}, 1 \times 10^{-4}\}$ | $1 \times 10^{-4}$ |
|  | Effective rank lr | $\{1 \times 10^{-3}, 1 \times 10^{-4}, 1 \times 10^{-5}\}$ | $1 \times 10^{-7}$ |
| $L2$-ER | Learning rate | $\{1 \times 10^{-4}\}$ | $1 \times 10^{-4}$ |
|  | Effective rank lr | $\{1 \times 10^{-5}, 1 \times 10^{-6}, 1 \times 10^{-7}\}$ | $1 \times 10^{-6}$ |
|  | Weight decay | $\{1 \times 10^{-3}, 1 \times 10^{-4}, 1 \times 10^{-5}\}$ | $1 \times 10^{-3}$ |
| CBP+$L2$ | Learning rate | $\{1 \times 10^{-4}\}$ | $1 \times 10^{-4}$ |
|  | Replacement rate | $\{1 \times 10^{-5}, 1 \times 10^{-6}, 1 \times 10^{-7}\}$ | $1 \times 10^{-7}$ |
|  | Weight decay | $\{1 \times 10^{-3}, 1 \times 10^{-4}, 1 \times 10^{-5}\}$ | – |
| $L2$ | Learning rate | $\{1 \times 10^{-2}, 1 \times 10^{-3}, 1 \times 10^{-4}\}$ | $1 \times 10^{-4}$ |
|  | Weight decay | $\{1 \times 10^{-3}, 1 \times 10^{-4}, 1 \times 10^{-5}\}$ | $1 \times 10^{-3}$ |

Table 7: Hyperparameter sweeps and selected best values for Slippery Ant across all algorithms.

### K.3  EXPERIMENTS

We swept the environment over 5 seeds for all hyperparameters in Table 7. Following the architecture of CBP in Dohare et al. (2024), we use CBP together with $L2$ normalization instead of just CBP. With the additional sweep on weight decays, we fixed the learning rate of CBP constant. The best hyperparameters are reported in Table 7.

| Hyperparam Name | Default | Description |
|---|---|---|
| env | slippery_ant | environment name |
| num_envs | 1 | number of parallel environments |
| gamma | 0.99 | discount factor |
| num_steps | 2048 | steps per rollout |
| update_epochs | 10 | number of optimization epochs per update |
| num_minibatches | 128 | number of minibatches per epoch |
| activation | relu | activation function: {relu, tanh} |
| optimizer | adam | optimizer: {adam, sgd, muon} |
| lr | [2.5e-4] | learning rate(s) |
| lambda0 | [0.95] | GAE parameter $\lambda$ |
| vf_coeff | [1] | value function loss coefficient |
| weight_decay | 0.0 | $L2$ weight decay |
| beta_1 | 0.9 | Adam $\beta_1$ coefficient |
| beta_2 | 0.999 | Adam $\beta_2$ coefficient |
| **continual backpropagation** | | |
| cont_backprop | False | enable CBP |
| replacement_rate | $1 \times 10^{-4}$ | CBP replacement probability per step |
| decay_rate | 0.99 | exponential decay for CBP statistics |
| maturity_threshold | $1 \times 10^4$ | steps before a unit is considered mature |
| **effective rank** | | |
| er | False | enable ER regularization |
| er_lr | [0.01] | ER learning rate |
| er_batch | 128 | batch size for ER computation |
| er_step | 1 | ER update frequency |
| **hessian computation** | | |
| compute_hessian_init | False | compute Hessian at initialization |
| compute_hessian_end | False | compute Hessian at the end |
| compute_hessian_size | 2000 | number of samples for Hessian computation |
| compute_hessian_interval | 1 | interval (in tasks) between Hessian runs |
| **PPO** | | |
| hidden_size | 256 | size of hidden layers |
| total_steps | $5 \times 10^6$ | total number of training steps |
| entropy_coeff | 0.01 | entropy regularization coefficient |
| clip_eps | 0.2 | PPO clipping parameter |
| max_grad_norm | $1 \times 10^9$ | gradient clipping threshold |
| anneal_lr | False | anneal learning rate over training |
| **evaluation** | | |
| steps_log_freq | 4 | logging frequency in steps |
| update_log_freq | 8 | logging frequency in updates |
| save_checkpoints | False | save checkpoints during training |
| save_runner_state | False | save final runner state |
| seed | 2020 | random seed |
| n_seeds | 5 | number of random seeds |
| platform | gpu | {cpu, gpu} |
| debug | False | enable debug mode |
| show_discounted | False | show discounted returns in logs |
| study_name | batch_ppo_test | experiment name |
| change_every | $2 \times 10^6$ | steps between environment changes |
| friction_seed | 0 | seed for friction schedule |

Table 8: Non-stationary policy default hyperparameters

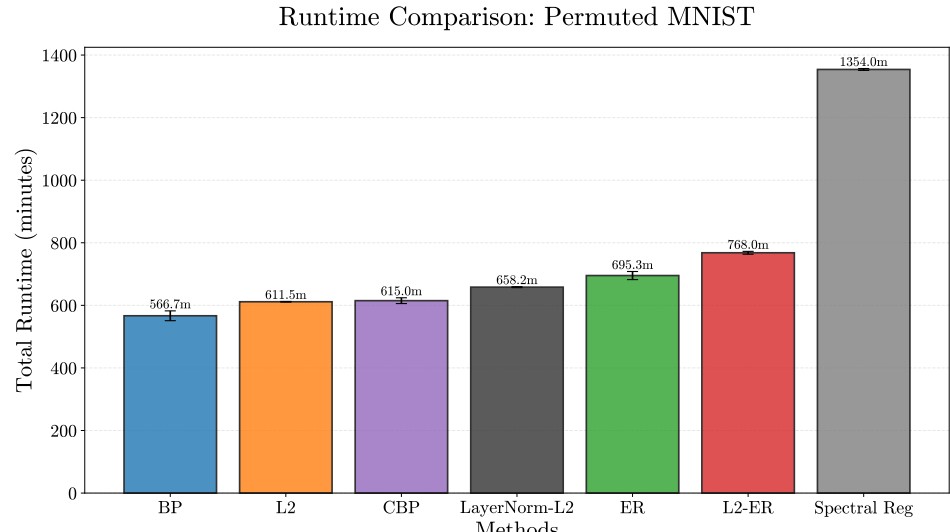

Figure 15: Runtime in Permuted MNIST Across all methods

## L  RUNTIME

## M  LLM Usage

Large Language Models were used to aid in the writing and editing of the manuscript. They are not used to form ideas, design experiments, or construct writing from scratch. The authors take full responsibility for the content of this manuscript and have ensured that LLM usage adhere to ethical guidelines.

