# OpenReview forum: "Spectral Collapse Drives Loss of Plasticity in Deep Continual Learning"
_ICLR.cc/2026/Conference — Submitted to ICLR 2026_

### Official Review · Reviewer_c59C · 2025-10-25

**Soundness:** 3
**Presentation:** 2
**Contribution:** 2
**Rating:** 4
**Confidence:** 3

**Summary:**

The paper investigates the loss of plasticity in continual learning from the perspective of the Hessian eigenspectrum and proposes a principled necessary condition for successful continual training.  The paper proposes that neural networks lose the ability to learn new tasks because their Hessian spectra collapse at new-task initialization: most curvature directions vanish, and gradient descent becomes ineffective in a continual learning setup. The paper formalizes this by introducing τ-trainability (a task-dependent requirement on Hessian rank) and proving that inactive (“dead”) neurons reduce Hessian rank and thus limit trainability — in particular, Theorem 6.2 relates the number of dead neurons to an upper bound on Hessian rank, and Theorem 6.5 gives a probabilistic bound showing that if many neurons are likely to persist dead across a task shift then the probability the next task is trainable is small.

Methodologically, the paper combines (i) spectral diagnostics, (ii) analytic bounds, and (iii) a simple, practical mitigation. The Hessian spectrum is estimated using stochastic Lanczos quadrature to empirically track spectral collapse across supervised and RL continual benchmarks; these experiments reveal a clear association between low ε-rank (collapsed spectrum) and failed training. Building on a Kronecker-factored view of curvature, the paper proposes a regularizer (L2-ER method) that maintains high effective feature rank and preserves curvature in the directions that matter. Empirically, L2-ER (and related curvature-preserving interventions) prevent spectral collapse and retain plasticity across Permuted MNIST, Incremental CIFAR, Continual ImageNet, and RL tasks, thereby confirming the theoretical connection to practice.

**Strengths:**

The paper investigates the still poorly understood problem of plasticity loss in neural networks and proposes an interesting and principled view on the relation between Hessian effective rank and network trainability.
In particular, I find the following strengths of the paper:

1. The paper rigorously connects “dead” ReLU neurons to reductions in Hessian rank (e.g., Theorem 6.2) rather than relying on hand-wavy intuition. This provides a concrete mechanism explaining why spectral collapse hinders optimization: inactive neurons eliminate curvature directions, thereby reducing the effective second-order information that gradient descent can utilize.
2. Lemma 6.3 (and the associated per-neuron margin condition) turns the abstract idea of dead neuron persistence into a simple geometric inequality that links margin, weight norm, and worst-case dataset shift (Hausdorff). That makes the theory falsifiable and directly measurable in practice
3. The proposed method, the L2-ER method, seems to be well principled (yet it should be evaluated more rigorously - see Weaknesses).

**Weaknesses:**

1. The paper should differentiate the contributions from the previous work [1, 2], with emphasis on [1], which also investigates the impact of the network's Hessian rank on the trainability in a continual setup. In the current form, the novelty of the paper is hard to assess.
2. The number of baseline methods used in the paper and the evaluation setup is unsatisfactory. I recommend
a) investigating how the proposed method L2-ER compares to Spectral Regularization [2] and one more such as Shrink and Perturb [4] or ReDO [3],
b) provide an additional investigation in reinforcement learning setups, where the plasticity issues often are more severe than in supervised ones.
3. The problem illustration in the introduction is confusing because there is no mention of how curvature-regularized methods work. Nevertheless, they are used in Figure 1 with no proper description.

[1] Lewandowski, A., Tanaka, H., Schuurmans, D., & Machado, M. C. (2023, November 30). Curvature Explains Loss of Plasticity. http://arxiv.org/abs/2312.00246
[2] Alex Lewandowski et al. (2024, June 10). Learning Continually by Spectral Regularization. http://arxiv.org/abs/2406.06811
[3] Sokar, G., Agarwal, R., Castro, P. S., & Evci, U. (2023, June 13). The Dormant Neuron Phenomenon in Deep Reinforcement Learning. http://arxiv.org/abs/2302.12902
[4] Ash, J. T., & Adams, R. P. (2020, December 31). On Warm-Starting Neural Network Training. http://arxiv.org/abs/1910.08475

**Questions:**

I don't understand the right plot of Fig. 1, in particular, why the bottom right plot looks like it has not converged?

---

> ### Author Response · Authors · 2025-11-20
> **Rebuttal part 1**
>
> We thank the reviewer for their feedback. We were glad that the reviewer found our contribution interesting and found our connection between ReLU neurons to reductions in Hessian rank useful.
> We address the weaknesses and questions below:
>
> ### Weaknesses:
>
> > - The paper should differentiate the contributions from the previous work [1, 2], with emphasis on [1], which also investigates the impact of the network's Hessian rank on the trainability in a continual setup. In the current form, the novelty of the paper is hard to assess.
>
> [1] establishes an empirical correlation between declining curvature and plasticity loss, measured via stable rank on a partial blockwise Hessian and supported by a proximity-to-init regularizer. Our paper extends this line of work into a precise theory to mechanism to remedy pipeline. (i) We introduce τ-trainability, a necessary condition for learning a new task: the full-Hessian ε-rank at task start must exceed a task-dependent threshold. This elevates “curvature matters” from association to a testable success criterion that predicts when learning will fail. (ii) We give a theoretical analysis: dead-neuron counts upper-bound Hessian rank (Theorem 6.2) and we provide a probabilistic bound showing how dead units persist across task shifts, yielding quantitative limits on τ-trainability. (iii) We diagnose spectral collapse (most eigenvalues near zero) as the proximate cause of failed adaptation, and validate the ε-rank <=> accuracy link at scale (e.g., Continual ImageNet) alongside landscape/spectrum visualizations. In summary, our paper complements [1] by building the theory and extending the empirics to the full Hessian spectrum.
>
> [2] stabilizes training by constraining layer spectral norms, but our work explains *why and when* such constraints help—and where they are insufficient. Through a K-FAC/Generalized Gauss–Newton analysis we show that continual failure is driven by Hessian spectral collapse at task switch; trainability requires preserving ε-rank (many non-negligible curvature directions), not just keeping the largest singular value in check. This lens clarifies why spectral regularization can work: by curbing norm blow-up it can *indirectly* slow rank collapse (e.g., fewer dead neurons, less gradient collinearity), which preserves some curvature directions and thus plasticity. But norm control alone doesn’t target the rank that our theory identifies as essential. Accordingly, we introduce L2-ER, a targeted regularizer derived from the K-FAC/GGN decomposition that directly preserves Hessian rank by increasing the effective ranks of the feature and gradient covariances. In short, our analysis both explains the efficacy of [2] and goes beyond it with a mechanism-aligned objective that safeguards the multiplicity of useful curvature directions required for continual learning.
>
> As a consequence of our theory, we find that both L2 *and* ER are needed to preserve plasticity; each loses plasticity on their own but preserves plasticity when applied together.
>
> > - The number of baseline methods used in the paper and the evaluation setup is unsatisfactory. I recommend a) investigating how the proposed method L2-ER compares to Spectral Regularization [2] and one more such as Shrink and Perturb [4] or ReDO [3], b) provide an additional investigation in reinforcement learning setups, where the plasticity issues often are more severe than in supervised ones.
>
> We have finished running layer normalization and spectral regularization baselines on permuted MNIST. These results are **already in the updated manuscript (Figure 3: top left, Figure 7 top)**. Layer normalisation with L2 still loses plasticity gradually. Spectral regularisation, while maintaining plasticity, runs much slower than our method due to power iteration at each gradient step (as shown in Figure 15). We are currently running these baselines on the rest of our experiments and will provide the results as the experiments finish.

---

> > ### Author Response · Authors · 2025-11-20
> > **Rebuttals part 2**
> >
> > > - The problem illustration in the introduction is confusing because there is no mention of how curvature-regularized methods work. Nevertheless, they are used in Figure 1 with no proper description.
> > >
> > Thank you for pointing out the ambiguity, we will surface the definition from Appendix C in the main text/caption. Specifically, curvature-GD minimizes
> >
> > $$(F\_t(\\theta)=L\_t(\\theta)+\\alpha\_t,R\_{\\text{curve}}(\\theta;L\_t)+\frac{\beta}{2}|\theta|*2^2),$$
> >
> > with $(R*{\text{curve}}(\theta;L\_t)=-\sum\_i \log(\lambda\_i(\theta;L\_t)^2+\epsilon))$, where $(\lambda\_i)$ are the Hessian eigenvalues; updates use the same step size and differ only by the added gradient term . *We have added additional information to the caption of Figure 1 and point to Appendix C for full details.*
> >
> > Questions:
> > >- I don't understand the right plot of Fig. 1, in particular, why the bottom right plot looks like it has not converged?
> > >
> >
> > Both methods are run for the same number of steps. GD does not converge because it is operating on a poorly conditioned subspace, so each gradient step makes little progress towards the minimum. We reiterate that Figure 1 is illustrative; the evidence for our claims comes from the Hessian eigenspectra measured at task boundaries (Fig. 2 right panel, and Figs 10,14,15), where standard BP shows progressive spectral narrowing/collapse while L2-ER preserves a broader spectrum. We’ll make this toy-vs-measurement distinction explicit in the caption and main text.
> >
> > Thank you again for taking the time to review the paper and providing helpful feedback! Do the above actions address your concerns with the paper? If not, what further clarification or modifications could we make to improve it?
> >
> > #### References
> > [1] Lewandowski, A., Tanaka, H., Schuurmans, D., & Machado, M. C. (2023, November 30). Curvature Explains Loss of Plasticity. http://arxiv.org/abs/2312.00246
> >
> > [2] Alex Lewandowski et al. (2024, June 10). Learning Continually by Spectral Regularization. http://arxiv.org/abs/2406.06811
> >
> > [3] Sokar, G., Agarwal, R., Castro, P. S., & Evci, U. (2023, June 13). The Dormant Neuron Phenomenon in Deep Reinforcement Learning. http://arxiv.org/abs/2302.12902
> >
> > [4] Ash, J. T., & Adams, R. P. (2020, December 31). On Warm-Starting Neural Network Training. http://arxiv.org/abs/1910.08475

---

> > > ### Comment · Reviewer_c59C · 2025-11-22
> > >
> > > Can the authors elaborate on the hyperparameters used in the new baselines, especially the spectral regularization coefficient?

---

> ### Author Response · Authors · 2025-11-22
>
> We adopted the implementation provided by 'Learning Continually by Spectral Regularization' with hyperparams provided in Table 1, appendix H.3 of the updated Manuscript. We additionally sweep the spectral strength hyperparam and reported the best choice in Table 2, appendix H.4 (best hyperparams adopted: k=2, target=2.0, spectral_strength=0.001, num_iter=10). We have updated Table 2 to also include Layernorm Hyperparameters.

---

### Official Review · Reviewer_4txZ · 2025-10-31

**Soundness:** 2
**Presentation:** 2
**Contribution:** 2
**Rating:** 2
**Confidence:** 3

**Summary:**

The paper argues that loss of plasticity in continual learning is driven by Hessian Spectral Collapse at new-task initialization. It introduces $\tau$-trainability and upper-bounds its probability by the $\epsilon$-rank and further by the number of dead neurons in a 2-layer ReLU MLP setting. The authors propose a regularization which combines an effective feature rank penalty with L2 weight decay.

**Strengths:**

The paper addresses a topic of continually growing importance. The problem of loss of plasticity in Continual Learning is well-motivated and clearly formulated, and Sections 1–5 are very well written. The paper frames prior work on loss of plasticity through the lens of Hessian spectral collapse. The authors also evaluate on standard Continual Learning benchmarks and conduct a substantial hyper-parameter sweep.

**Weaknesses:**

Section 7, on mitigating spectral collapse, is poorly written. Key quantities are not introduced or defined, and the approximations are insufficiently justified. Equation 5 contains typos and the approximation error in equation 6 is particularly unclear.

The experiments also lack comparisons to related work on mitigating loss of plasticity (e.g., [1]); they evaluate the proposed method only against a single Continual Learning baseline and standard backprop. Additional potential weaknesses are noted in the questions.


[1] Alex Lewandowski, MichaĹ Bortkiewicz, Saurabh Kumar, Dale Schuurmans, Mateusz Ostaszewski,
Marlos C Machado, et al. Learning continually by spectral regularization. arXiv preprint
arXiv:2406.06811, 2024a.

**Questions:**

- Why restrict the theoretical analysis to a two-layer MLP? What are your empirical observations on how wider/deeper networks affect trainability?
- Why is the analysis conducted using the notion of $\epsilon$-rank rather than the effective rank used in the regularization term? How are the two related?

I would be happy to raise my score if the authors address the fundamental issues noted in the weaknesses.

---

> ### Author Response · Authors · 2025-11-20
> **Rebuttal part 1**
>
> We thank the reviewer for their comments. We were glad the reviewer found our problem setting important and that the majority of the paper was well written. We address the weakness and questions below:
>
> Weaknesses:
> > Section 7, on mitigating spectral collapse, is poorly written. Key quantities are not introduced or defined, and the approximations are insufficiently justified. Equation 5 contains typos and the approximation error in equation 6 is particularly unclear.
>
> Thank you for the feedback. We have **integrated changes into section 7** to bring it to the same standard as the earlier sections. In particular, we have re-defined the GNN decomoposition and Kronecker factorization with more consistent notation. We have also explained the error derivation in Appendix D of the manuscript. In particular:
>
> For a given layer, $\\mathbf{W}\_l \\in \\mathbb{R}^{d\_{in} \\times d\_{out}}$, with activations into the layer $\\mathbf{a}\_l \\in \\mathbb{R}^{d\_{in}}$ and gradient outputs $g \\in \\mathbb{R}^{d\_{out}}$. The respective covariances are $\\mathbf{a}\_l\\mathbf{a}\_l^\\top \\in \\mathbb{R}^{d\_{in} \\times d\_{in}}$ and $g\_l g\_l^\\top \\in \\mathbb{R}^{d\_{out} \\times d\_{out}}$. The resulting KFAC block is of size $(d\_{out}d\_{in}) \\times (d\_{out}d\_{in})$. The per-entry error, on the $n$th sample, can be obtained by indexing over Kronecker matrix pairs with row-index $(i, \\alpha)$ and column index $(j, \\beta)$. We use $\\cdot\_{l,[i,\*]}$ to denote the $i$th row of $(\\cdot)$ in layer $l$, and $\\cdot\_{l,[\*,j]}$ for the $j$th column. Flattening over the Kronecker product, we obtain the following for a given entry in layer $l$:
>
> $$
> (\\mathbf{H}\_{GGN})\_{l,[i,\\alpha],[j,\\beta]} = \\frac{1}{N} \\sum\_{n=1}^{N} \\sum\_{o=1}^{O} \\mathbf{a}\_{l[i,\*]}^n \\mathbf{a}\_{l[\*,j]}^n \\mathbf{g}\_{l[\\alpha,\*]}^{n,o} \\mathbf{g}\_{l[\*,\\beta]}^{n,o} = \\overline{\\mathbf{a}\_{l[i,\*]}\\mathbf{a}\_{l[\*,j]}\\mathbf{g}\_{l[\\alpha,\*]}\\mathbf{g}\_{l[\*,\\beta]}} \\tag{48}
> $$
>
> $$
> (\\hat{\\mathbf{H}}\_{GGN})\_{l,[i,\\alpha],[j,\\beta]} = \\left( \\frac{1}{N} \\sum\_{m=1}^{N} \\mathbf{a}\_{l[i,\*]}^n \\mathbf{a}\_{l[\*,j]}^n \\right) \\cdot \\left( \\frac{1}{N} \\sum\_{n=1}^{N} \\sum\_{o=1}^{O} \\mathbf{g}\_{l[\\alpha,\*]}^{m,o} \\mathbf{g}\_{l[\*,\\beta]}^{m,o} \\right) = \\overline{\\mathbf{a}\_{l[i,\*]}\\mathbf{a}\_{l[\*,j]}} \\cdot \\overline{\\mathbf{g}\_{l[\\alpha,\*]}\\mathbf{g}\_{l[\*,\\beta]}} \\tag{49}
> $$
>
> where we use $\\overline{\\cdot}$ to denote the empirical sample average. In this way, we can compute the error as the difference between the average of products and the product of averages; for a fixed index quadruple $(i, j, \\alpha, \\beta)$, the KFAC error of the $((i, \\alpha), (j, \\beta))$th-entry is:
>
> $$
> \\Delta\_{[i,\\alpha],[j,\\beta]} = (\\mathbf{H}\_{GGN} - \\hat{\\mathbf{H}}\_{GGN})\_{[i,\\alpha],[j,\\beta]} = \\overline{\\mathbf{a}\_{l[i,\*]}\\mathbf{a}\_{l[\*,j]}\\mathbf{g}\_{l[\\alpha,\*]}\\mathbf{g}\_{l[\*,\\beta]}} - \\overline{\\mathbf{a}\_{l[i,\*]}\\mathbf{a}\_{l[\*,j]}} \\cdot \\overline{\\mathbf{g}\_{l[\\alpha,\*]}\\mathbf{g}\_{l[\*,\\beta]}} \\tag{50}
> $$
>
> Which is small if $\\mathbf{a}\_l\\mathbf{a}\_l^\\top$ and  $\\mathbf{g}\_l \\mathbf{g}\_l^\\top$ are independent which appears to hold in practice [2].
>
> > The experiments also lack comparisons to related work on mitigating loss of plasticity (e.g., [1]); they evaluate the proposed method only against a single Continual Learning baseline and standard backprop. Additional potential weaknesses are noted in the questions.
>
> We have finished running layer normalization and spectral regularization baselines on permuted MNIST. These results are already in the updated manuscript (Figure 3: top left, Figure 7 top). Layer normalisation with L2 still loses plasticity gradually. Spectral regularisation, while maintaining plasticity, runs much slower than our method due to power iteration at each gradient step (as shown in FIgure 15). We are currently running these baselines on the rest of our experiments and will provide the results as the experiments finish.

---

> > ### Author Response · Authors · 2025-11-20
> > **Rebuttal part 2**
> >
> > Questions:
> > > Why restrict the theoretical analysis to a two-layer MLP? What are your empirical observations on how wider/deeper networks affect trainability?
> >
> > We focus on a two-layer MLP as the minimal nonlinear setting that captures the mechanisms of interest—ReLU gating, dead-neuron–induced rank limits, and Hessian spectral collapse, without the technical overhead of depth. This choice makes the analysis clean and testable. The resulting bounds and intuitions are layer-local (via per-layer Jacobian/feature covariances), and extend naturally to deeper networks through standard block/diagonal factorization (e.g., K-FAC/GAUSS–Newton style), which our experiments on deeper models empirically corroborate.
> >
> > Our experiments are indeed run on a variety of networks. Permuted MNSIT is run with a small feedforward MLP (appendix H.1). Continual imagenet is run with a convolutional neural network (appendix I.1), and incremental CIFAR was run on a ResNet (appendix J.1), which includes skip connections. We also use SGD for supervised learning task and Adam for the RL task since they are two of the most popular optimizers used in the community and find that both lead to loss of plasticity. Our existing analysis suggests that optimizers that improve the conditioning of the Hessian, like AdamW through L2 reg, would improve plasticity since it targets the same underlying mechanism.
> >
> > > Why is the analysis conducted using the notion of $\epsilon$-rank rather than the effective rank used in the regularization term? How are the two related?
> >
> > Because τ-trainability is a *curvature* condition: learning succeeds only if there are sufficiently many positive-curvature directions which is why we anaylze the Hessian. Directly accessing the full Hessian is intractable, so we work through the Gauss–Newton/KFAC factorization, where the per-layer GGN decomposes into a Kronecker product of feature covariance and output-gradient covariance; consequently, the rank of $H\_{GGN}$ is monotone in the ranks of these two covariances (by $rank(A\otimes B)=rank(A)\cdot rank(B))$. This gives a concrete handle we can optimize: we maximise the effective rank (ER) of features to raise rank of the input covariance, thereby raising $rank(H_{GGN})$, which in turn raises $rank(H)$ under a mild L2 term. Formally, Theorem B.3 / Corollary B.4 show there exists an L2 strength such that increasing $rank(H_{GGN})$ implies increasing $rank(H)$, i.e., positive eigenvalues of $H_{GGN}$ remain positive in the L2-regularized Hessian. Thus, analyzing $\epsilon$-$rank(H)$ (for τ-trainability) and optimizing ER + L2 (for intervention) are two sides of the same pipeline: $\epsilon$-rank is the right object for the success criterion, while ER is the practical proxy via the GGN/Kronecker, structure that **provably transmits curvature into H**.
> >
> >
> > Thank you again for taking the time to review the paper and providing helpful feedback! Do the above actions address your concerns with the paper? If not, what further clarification or modifications could we make to improve it?
> >
> > #### References
> > [1] Alex Lewandowski, MichaĹ Bortkiewicz, Saurabh Kumar, Dale Schuurmans, Mateusz Ostaszewski, Marlos C Machado, et al. Learning continually by spectral regularization. arXiv preprint arXiv:2406.06811, 2024a.
> >
> > [2] Martens, James, and Roger Grosse. "Optimizing neural networks with kronecker-factored approximate curvature." International conference on machine learning. PMLR, 2015.

---

> > > ### Comment · Reviewer_4txZ · 2025-11-25
> > >
> > > Thank you for the clarifications.
> > >
> > > From Corollary B.4, the minimal L2 penalty to guarantee non-decreasing rank is at least $||R||_2 - \lambda^{+}_{min}(G_{GGN})$ . This condition seems to hold trivially for the two-layer Relu MLP of section 6. How does this bound look like for a more general class of networks, in particular how does it behave roughly as a function of the feature dimension?
> > >
> > > My lingering worry is that there is a large gap between the assumptions loosely justified in theory and the actual networks this procedure is applied to. In particular, the joint gaussianity assumption(line 393) and deep linear networks assumptions(395) are very restrictive assumptions to justify the KFAC approximation and the resulting KFAC error might be far too big to ignore in practice.

---

> > > > ### Author Response · Authors · 2025-11-26
> > > >
> > > > We thank the reviewer for these questions. We address (i) the scaling of the L2 bound in Corollary B.4 beyond the 2-layer MLP, and (ii) the practical validity of the KFAC approximation.
> > > >
> > > > (i) On the bound $2q \\ge ||R||\_2 - \\lambda\_{\\min}\^+(H\_{\\text{GGN}})$ and dependence on network size.
> > > >
> > > > Corollary B.4 is stated for a general $L$-layer MLP and does not rely on the specific 2-layer ReLU example in Section 6. The residual term $||R||\_2$ is controlled in Theorem B.3, where we show:
> > > >
> > > > $$||R(\\theta\^0\_{\\tau+1})||\_2 \\le \\sqrt{O} \\Gamma(\\theta\_\\tau) \\Big( \\sup\_{x \\in \\mathrm{supp}(X\_\\tau)} ||\\nabla\_z \\ell(z,y)||\_2 + L\_f L\_g d(X,X')\\Big)$$
> > > >
> > > > Here, $O$ is the **output dimension**, and $\Gamma(\theta)$ is the smoothness constant of the logit map in parameter space. Crucially, the bound scales with $\sqrt{O}$ rather than the hidden feature dimension. While the Lipschitz constants ($L_f, L_g$) and smoothness ($\Gamma$) could theoretically grow with width, standard practices we employ (Kaiming initialization and L2 regularization) keep parameter norms bounded. This implies $\Gamma(\theta) \le C$ for some architecture-dependent constant $C$ that does not explode with width.
> > > >
> > > > Consequently, the required penalty $q\_{\\min} \\approx \\frac{1}{2}(||R||\_2 - \\lambda\_{\\min}\^+(H\_{\\text{GGN}}))$ remains manageable even for wide networks, provided the representations do not collapse (which maintains $\lambda_{\min}^+(H_{\text{GGN}})$). While the 2-layer MLP in Section 6 serves as an illustrative case, the bound confirms that a non-vacuous regime exists for general deep networks where increasing $\text{rank}(H_{\text{GGN}})$ guarantees non-decreasing rank of the true Hessian.
> > > >
> > > > (ii) On the KFAC approximation assumptions.
> > > >
> > > > We agree that the joint Gaussianity and deep-linear assumptions are restrictive. However, we employ them in the standard way found in the literature [1,2,3]: to motivate the tractable Kronecker-factored form of the regularizer. Empirically, KFAC has also been found to capture useful curvature information in CNNs [4], RNNs [5] and vision-transformers [6]. To the reviewer's point, [7] attempts to reduce the KFAC approximation error by re-introducing off-diagonal block interactions (which KFAC ignores) and conclude that "incorporating the layer interactions... does not really improve the performance", suggesting that it is indeed "safe to discard the off diagonal blocks".
> > > >
> > > > Ultimately, the validity of this approximation is an empirical question: does regularizing the KFAC proxy actually improve the geometry of the true loss landscape? Our experiments answer this affirmatively. We explicitly measure the eigenspectrum of the **true Hessian** (using stochastic Lanczos quadrature, not KFAC approximations) across diverse, non-linear architectures like ResNet-18 and ConvNets. In all cases, we observe that increasing the effective rank of the KFAC proxy directly prevents spectral collapse in the true Hessian. This confirms that, despite the theoretical approximation gap, the KFAC regularizer aligns sufficiently well with the true curvature directions to preserve plasticity in practice.
> > > >
> > > > ---
> > > >
> > > > Thank you again for the questions. Do the above answers address your concerns with the validity of our method? If not, what further clarifications can we provide?
> > > >
> > > > # References
> > > > [1] Martens, James, and Roger Grosse. "Optimizing neural networks with kronecker-factored approximate curvature." International conference on machine learning. PMLR, 2015.
> > > >
> > > > [2] Dangel, Felix, et al. "Kronecker-factored Approximate Curvature (KFAC) From Scratch." arXiv preprint arXiv:2507.05127 (2025).
> > > >
> > > > [3] Ba, Jimmy, Roger Grosse, and James Martens. "Distributed second-order optimization using kronecker-factored approximations." International conference on learning representations. 2017.
> > > >
> > > > [4] Grosse, Roger, and James Martens. "A kronecker-factored approximate fisher matrix for convolution layers." International Conference on Machine Learning. PMLR, 2016.
> > > >
> > > > [5] Martens, James, Jimmy Ba, and Matt Johnson. "Kronecker-factored curvature approximations for recurrent neural networks." International Conference on Learning Representations. 2018.
> > > >
> > > > [6] Eschenhagen, Runa, et al. "Kronecker-factored approximate curvature for modern neural network architectures." Advances in Neural Information Processing Systems 36 (2023): 33624-33655.
> > > >
> > > > [7] Koroko, Abdoulaye, et al. "Analysis and comparison of two-level KFAC methods for training deep neural networks." Optimization Methods and Software 40.1 (2025): 96-128.

---

### Official Review · Reviewer_b6hD · 2025-11-01

**Soundness:** 3
**Presentation:** 3
**Contribution:** 1
**Rating:** 2
**Confidence:** 4

**Summary:**

This paper aims to explain the loss of plasticity through the collapse of the Hessian eigenspectrum and thereby theoretically connects known drivers of plasticity loss, dormant neurons and effective rank. Based on this theoretical understanding, the paper proposes L2-ER regularization, which enables the model to maintain a stable Hessian spectrum during training.

**Strengths:**

The paper provides useful theoretical understanding on loss of trainability. The theory connects dormant neurons, effective rank, and loss curvature, which enrich understanding of plasticity loss.

**Weaknesses:**

* The method lacks novelty. The proposed L2-ER regularization is a simple combination of existing techniques — effective rank regularization [1] and L2 regularization.
* The theoretical results are incremental. It is known that loss curvature is deeply connected to loss plasticity [2, 3]. I acknowledge its novelty, but it is still incremental.
* The theoretical analysis could be further elaborated using observations from previous work. For example, [2] found that the sharpness of loss curvature correlates with the loss of plasticity, especially in the absence of dormant neurons.
* The experiments are insufficient to validate the effectiveness of the proposed method. Additional reinforcement learning experiments could be conducted in more environments.
* The idea that Figure 1 aims to convey is ambiguous. The authors mention that GD exhibits unstable learning dynamics (zig-zagging), while curvature-regularized GD remains stable. However, it is unclear which factor causes this difference. Is it due to different learning algorithms (GD vs. curvature GD), or to different starting points (valley vs. bowl)? In Task 2, the orange and blue lines start from different points and are minimized using different algorithms. Therefore, it is unclear which factor is responsible for the observed optimization stability. In addition, the starting point of GD is much closer to the valley, while that of curvature GD is much closer to the bowl. This might affect the optimization. Would the result remain the same if the loss landscape of Task 2 were vertically flipped?

[1] Kumar, Aviral, et al. "Implicit under-parameterization inhibits data-efficient deep reinforcement learning." arXiv preprint arXiv:2010.14498 (2020).
[2] Lyle, Clare, et al. "Understanding plasticity in neural networks." International Conference on Machine Learning. PMLR, 2023.
[3] Lewandowski, Alex, et al. "Directions of curvature as an explanation for loss of plasticity." arXiv preprint arXiv:2312.00246 (2023).

**Questions:**

This point is described in the weakness section.

---

> ### Author Response · Authors · 2025-11-20
> **Rebuttal part 1**
>
> We thank the reviewer for their comments. We were glad the reviewer found our understanding on loss of trainability useful.
>
> We address the questions and weaknesses below:
>
> Weaknesses:
> > - The method lacks novelty. The proposed L2-ER regularization is a simple combination of existing techniques — effective rank regularization [1] and L2 regularization.
>
> We thank the reviewer for pointing out [1], which **we have already added to our related work and discussion of ER in section 7**, noting that they explore the single task RL setting and not continual learning.
>
> We believe the beauty of our analysis lies in the simplicity of the resulting regularizer. Our novelty primarily comes from the theory connecting dormant neurons, effective rank, and loss curvature, which enriches the understanding of plasticity loss. As a consequence, we find that both L2 *and* ER are needed to preserve plasticity; each loses plasticity on their own but preserves plasticity when applied together. Our analysis in Theorem B.3 and Corollary B.4 explains why both are needed. We believe it is of interest to the broader community when well-known methods interact in unexpected ways.
>
> > - The theoretical results are incremental. It is known that loss of curvature is deeply connected to loss plasticity [2, 3]. I acknowledge its novelty, but it is still incremental.
>
> We agree that [2] documents a correlation between curvature loss and plasticity loss, measured via stable rank of a partial, blockwise * Hessian and proposes a Wasserstein-initialization regularizer to keep weight distributions near their initial order statistics. But our contribution is not another correlation study: we develop theory that explains why these empirical patterns occur. Specifically, we (i) formalize τ-trainability as a necessary condition based on the full-Hessian ε-rank at new-task initialization and show ε-rank tightly predicts success at scale (e.g., Continual ImageNet), directly tying spectral collapse to failure; and (ii) prove a mechanistic link, dead neurons upper-bound Hessian rank and persist across task shifts. (iii) Grounded in a KFAC/GGN decomposition, we further explain how to prevent collapse: **L2-ER** raises curvature by increasing the rank of the Gauss–Newton (and hence the Hessian), providing a principled explanation for the gains of curvature-preserving interventions rather than a distributional heuristic.
>
> [3] provide a broad empirical analysis showing that plasticity loss is connected to curvature/landscape sharpness and optimizer/architecture choices (e.g., LayerNorm), and they highlight non-stationarity-induced instabilities; however, they do not provide a trainability criterion or a mechanism that explains when learning will fail. We go beyond correlation by explaining the phenomenon: τ-trainability gives a curvature-based success condition; Theorem 6.2 and the persistence bound explain how dead neurons limit curvature across tasks; and our KFAC/GGN-derived L2-ER explains why certain interventions (in combination) preserve plasticity—because they explicitly increase curvature direction* and prevent spectral collapse. Our theoretical results explain these known empirical results, and go beyond. For example, with our notion of task shift, and our probabilistic bound, our theory explains speculations from [4] that a network may fall into an ill-conditioned subspace of the loss landscape on the current task, but then be transported to a well-conditioned region after a lucky task shift.
>
>
> > - The theoretical analysis could be further elaborated using observations from previous work. For example, [2] found that the sharpness of loss curvature correlates with the loss of plasticity, especially in the absence of dormant neurons.
>
> We appreciate this connection and, in fact, we subsume and sharpen it:
>
> [2] observes that curvature tracks plasticity and discusses dead neurons among several factors, providing counter-examples where single proxies fail. Our framework does not rely on dead neuron counts aligning: by defining spectral collapse via ε-rank of the full Hessian, we capture flatness induced by other mechanisms (e.g., gradient collinearity, low feature rank [3], and linearization of ReLU units) that can occur even when dormant neurons are scarce, precisely the regime the reviewer notes. Thus, our theory appies to [2].
>
> We incorporate neuron pathologies explicitly—and bound their impact. Rather than using dormancy only as an empirical correlate, we prove that dead-neuron counts upper-bound Hessian rank (Theorem 6.2) and derive the probability that neurons stay dead across task changes, delivering quantitative predictions** for when curvature (and thus plasticity) must fail. This turns the observations from [2,3] into measurable bounds.

---

> > ### Author Response · Authors · 2025-11-20
> > **Rebuttal part 2**
> >
> > > - The experiments are insufficient to validate the effectiveness of the proposed method. Additional reinforcement learning experiments could be conducted in more environments.
> >
> > We have finished running layer normalization and spectral regularization baselines on permuted MNIST. These results are already in the updated manuscript (Figure 3: top left, Figure 7 top). Layer normalisation with L2 still loses plasticity gradually. Spectral regularisation, while maintaining plasticity, runs much slower than our method due to power iteration at each gradient step (as shown in FIgure 15). We are currently running these baselines on the rest of our experiments and will provide the results as the experiments finish.
> >
> > > - The idea that Figure 1 aims to convey is ambiguous. The authors mention that GD exhibits unstable learning dynamics (zig-zagging), while curvature-regularized GD remains stable. However, it is unclear which factor causes this difference. Is it due to different learning algorithms (GD vs. curvature GD), or to different starting points (valley vs. bowl)? In Task 2, the orange and blue lines start from different points and are minimized using different algorithms. Therefore, it is unclear which factor is responsible for the observed optimization stability. In addition, the starting point of GD is much closer to the valley, while that of curvature GD is much closer to the bowl. This might affect the optimization. Would the result remain the same if the loss landscape of Task 2 were vertically flipped?
> >
> > Thank you for flagging the ambiguity. First, we would like to clarify that Figure 1 is a toy meant only to convey intuition about conditioning. Both methods start Task 1 from the same parameters, use the same step size, and differ only by the curvature regularizer. Because of this difference, they converge to different Task-1 minima (a flat “valley” for GD vs. a well-conditioned “bowl” for the regularized method). Task 2 then begins from these converged points after an affine transformation of the landscape; the observed zig-zagging reflects the fact that GD now starts from an ill-conditioned subspace, not step-size tuning or other implementation differences. The full details of the implementation can be found in appendix C. Of course, it would be possible to construct a loss landscape where vanilla GD is not pathological. We reiterate that Figure 1 is illustrative; the evidence for our claims comes from the Hessian eigenspectra measured at task boundaries (Fig. 2 right panel, and Figures 10,14,15), where standard BP shows progressive spectral narrowing/collapse while L2-ER preserves a broader spectrum. We’ll make this toy-vs-measurement distinction explicit in the caption and main text.
> >
> > Thank you again for taking the time to review the paper and providing helpful feedback! Do the above actions address your concerns with the paper? If not, what further clarification or modifications could we make to improve it?
> >
> > ### References
> > [1] Kumar, Aviral, et al. "Implicit under-parameterization inhibits data-efficient deep reinforcement learning." arXiv preprint arXiv:2010.14498 (2020).
> >
> > [2] Lyle, Clare, et al. "Understanding plasticity in neural networks." International Conference on Machine Learning. PMLR, 2023.
> >
> > [3] Lewandowski, Alex, et al. "Directions of curvature as an explanation for loss of plasticity." arXiv preprint arXiv:2312.00246 (2023).
> >
> > [4] Lyle, Clare, et al. "Disentangling the causes of plasticity loss in neural networks." arXiv preprint arXiv:2402.18762 (2024).

---

### Official Review · Reviewer_TdDm · 2025-11-02

**Soundness:** 2
**Presentation:** 3
**Contribution:** 2
**Rating:** 4
**Confidence:** 4

**Summary:**

This submission studies the trainability of deep networks from the perspective of the Hessian and how curvature affects neural network optimization dynamics. The paper provides evidence that spectral collapse underlies some of the deep network's loss of plasticity/trainability effect. They use these insights to propose a regularizer that operates on the hidden units of the network, and find that this proves to be effective in combination with l2 regularization.

While the paper is interesting and provides some insight, there are some weaknesses in the experiments that I am not sure a rebuttal can address. That is, the experiments seem fairly disconnected from the rest of the paper, and somewhat undermines the larger thesis.

**Strengths:**

- Overall, the paper is well written. The main narrative presented is largely compelling, and the main thesis that spectral collapse underlies loss of plasticity is well supported by previous work.
- Theoretical analysis of trainability appears novel and relevant. It differs from previous work that focus on a particular perspective, such as weight resetting or regularizing. The authors also translate these insights into a regularizer.

**Weaknesses:**

- The empirical results are lacking in a few respects. First, there are some important baselines left out (such as various regularizers, and neural network architectures including activation functions and normalization layers)
- Connected to the experiments, there is little connecting the theoretical results to the experiments besides the proposed method. While there was substantial discussion regarding spectral collapse, it was never actually measured empirically.
- There are some clarity issues surrounding parts of the definitions and theorems (see below)

**Questions:**

- How do you account for curvature and Hessian changes due to changes in the data distribution on the new task?
- line 45: This introduction leaves out some previous work that has looked at second order effects on plasticity and found mixed results. While the related work goes into some detail, the introduction positions the submission as if it is the first to tackle this direction.
- I am confused by why the bowls and saddles in figure 1 and in appendix A are associated with spectral collapse. Is it right to interpret this low dimensional projection as evidence that the high dimensional manifold is actually flat and suffering from spectral collapse? Furthermore, I would be curious to know how these landscapes look at the end of the previous task. Does l2-er reach a local minimum?
- Definition 5.1 doesn't seem like a practical definition, because in practice the set of arg-maximizers can be a set of measure zero. Why not loosen this definition to include an error tolerance?
- Definition of trainable as "initialized with fewer than $m$ dead neurons" doesn't seem correct. For one thing, it doesnt respect architecture (where are the dead neurons located?) nor size (what proportion of units are dead?). For example, a very large architecture with $500$ dead neurons will likely be trainable, whereas a small network may not be.
- Definition 5.2: This definition does not seem operationalizable without specifying a reasonable threshold for each task. How does the threshold depend on the task, and on the network/optimizer?
- Theorem 6.2: Isnt it the case that $\rho_\tau = \text{rank}(H_\tau^{(0)})$? If so, then the inequality is always satisfied, and you have the vacuous bound $P(T_\tau = 1) < 1$
- line 380: Im confused as to why l2 regularization would increase the rank, since L2 regularization would regularize the eigenvalues towards zero.
- Missing related work: parseval regularization seems to capture a very similar same term, by orthogonalizing the weights.
- Why isnt the hessian rank measured in the experiments?
- Doesn't this claim: "Single metrics are insufficient." undermine the narrative surrounding spectral collapse as the unifying explanation?
- Missing layer normalization baseline: it is now common to use normalization, and this has also been found to greatly reduce loss of plasticity.
- The neural network architectures here are rather small and basic, lacking normalization and skip connections. There is also comparatively little analysis of how the optimizer may contribute to curvature, as compared to previous work.

### Minor Comments
- line 168: $\theta^*$ is missing a subscript

---

> ### Author Response · Authors · 2025-11-20
> **rebuttal part 1**
>
> We thank the reviewer for their feedback. We were glad to see that the reviewer found our analysis of trainability appears novel and relevant, and the paper well-written. We hope the following addresses their questions and concerns:
>
> ## Weaknesses:
> > - The empirical results are lacking in a few respects. First, there are some important baselines left out (such as various regularizers, and neural network architectures including activation functions and normalization layers)
>
> We have finished running layer normalization and spectral regularization baselines on permuted MNIST. These results are already in the updated manuscript (Figure 3: top left, Figure 7 top). Layer normalisation with L2 still loses plasticity gradually. Spectral regularisation, while maintaining plasticity, runs much slower than our method due to power iteration at each gradient step (as shown in FIgure 15). We are currently running baselines in the rest of our experiments and will also provide quantitative results as the experiments finish.
>
> > - Connected to the experiments, there is little connecting the theoretical results to the experiments besides the proposed method. While there was substantial discussion regarding spectral collapse, it was never actually measured empirically.
>
> We measure the Hessian eigenspectrum of all algorithms on all task initializations throughout our experiments; these can be found in figures 10, 12 and 13. Figure 2 (right panels) shows the eigenspectrum before and after collapse on continual imagenet.
>
> Measuring and understanding the spectral collapse in various tasks, architectures, and setups, **is an important part of the experiments we conduct**. These empirical findings prompted us to bring together current explanations of loss of plasticity under the framework of $\tau$-trainability. Then, in sections 7 and 8, we proposed a regularizer to mitigate spectral collapse, and empirically demonstrated the effectiveness of this regularizer.
>
> In particular, we argue that increasing the feature $\epsilon$-rank of the network implies an increase in the $\epsilon$-rank of the Gauss Newtown Matrix which implies an increase in the $\epsilon$-rank of the Hessian which we measure.
>
>
>
> ## Questions:
> > - How do you account for curvature and Hessian changes due to changes in the data distribution on the new task?
>
> We connect curvature to dormant neurons in Theorem 6.2. In 6.3-6.5, we quantify the data distribution shift via the Hausdorff distance between two training sets. This gives us the ability to reason about the upperbound of the curvature on new task initialization, which by definition poses an upperbound on the trainability of the new task.
>
>
> > line 45: This introduction leaves out some previous work that has looked at second order effects on plasticity and found mixed results. While the related work goes into some detail, the introduction positions the submission as if it is the first to tackle this direction.
>
> Thank you for bringing this to our attention. We acknowledge that previous papers[1,2] have discussed curvature relating to plasticity, but what we offer is a unifying theoretical lens into how curvature, Hessian rank and dead neurons are connected. **We have updated the introduction to include these works and distinguish our contributions**
>
> > - I am confused by why the bowls and saddles in figure 1 and in appendix A are associated with spectral collapse. Is it right to interpret this low dimensional projection as evidence that the high dimensional manifold is actually flat and suffering from spectral collapse? Furthermore, I would be curious to know how these landscapes look at the end of the previous task. Does l2-er reach a local minimum?
>
> The figure is a toy construction meant to illustrate how ill-conditioned subspaces (flat “canyons”) versus well-conditioned basins (bowls) lead to different dynamics. In particular, initialization in low-rank subspaces (canyons) can lead to poorer learning dynamics. The 2-D panels show this intuition; they are not the evidence for collapse. Our claims of spectral collapse come from quantitative Hessian measurements made at the beginning of each task using stochastic Lanczos quadrature. Under standard BP the spectrum narrows and can completely collapse as tasks progress, whereas L2-ER preserves a wider spectrum. These plots can be seen in Figures 10, 14 and 15. **We have also made the Hessian eigenspectrum distribution at the end of training available (Figure 11), which also exhibits a narrower distribution on the train set for BP and a wider distribution for plasticity preserving algorithms**.

---

> ### Author Response · Authors · 2025-11-20
> **Rebuttal part 2**
>
> > - Definition 5.1 doesn't seem like a practical definition, because in practice, the set of arg-maximizers can be a set of measure zero. Why not
> loosen this definition to include an error tolerance?
>
> Thank you for pointing this out. **We have modified  definition 5.1** to define successful training when
>
> $$
> \mathbb{I}(f\_\tau \left(\theta_\tau^{(K)}\right) \ge \sup_{\theta} f_\tau(\theta) - \upsilon),
> $$
> where $\upsilon > 0$ is an error-tolerence term.
>
>
> > - Definition of trainable as "initialized with fewer than
> >  dead neurons" doesn't seem correct. For one thing, it doesnt respect architecture (where are the dead neurons located?) nor size (what proportion of units are dead?). For example, a very large architecture with
> >  dead neurons will likely be trainable, whereas a small network may not be.
>
> Thank you for pointing this out! We have made the correction such that  the network is "initialized with at least $a_\tau$ active neurons".
>
> > - Definition 5.2: This definition does not seem operationalizable without specifying a reasonable threshold for each task. How does the threshold depend on the task, and on the network/optimizer?
>
> You are right in pointing out that the threshold is conditioned on the task, the network and the optimizer. Therefore, for a specific combination of task, network, and optimizer, we can measure the trainable threshold empirically. We provide examples of this in Figure 2 and in Appendix F. In particular, we examine the normalized $\epsilon$-rank, which is the portion of absolute eigenvalues above $\epsilon$. In Figure 2, suppose successful training means to achieve at least 80% accuracy on continual Imagenet (the task) with the Resnet architecture in specified in appendix H.1 (the network) and optimizer SGD (the optimizer), the network is said to be trainable if at least 0.5 (50%) of the eigenvalues are above 0.1 (where 0.1 is the $\epsilon$ in normalized $\epsilon$-rank). Figure 2 depicts this relationship. **We have adjusted definition 5.2 to make it clear that it is also optmizer dependent as well.**
>
>
> > - Theorem 6.2: Isnt it the case that
> > ? If so, then the inequality is always satisfied, and you have the vacuous bound
>
> $\rho_\tau \neq rank(H_\tau^{0})$. $\rho_\tau$ is the task and network dependent threshold for trainability as defined in 5.2. Therefore the bound is not vacuous.
>
> > - line 380: Im confused as to why l2 regularization would increase the rank, since L2 regularization would regularize the eigenvalues towards zero.
>
> It's true that addling L2 regularization penalizes the weight matrices from getting large, but this doesn't necessarily mean the eigenvalues of the Hessian would be regularized to zero. Perhaps the reviewer can point us to a reference?
>
> L2 regularization was initially introduced for deep ReLu networks to ensure that neurons don't get saturated [2] because of the large magnitude of the matrix weights. Our work argues that the core mechanism through which plasticity is maintained, in the language of optimization, is by ensuring a wide bulk of the spectrum which is achieved by combining L2 reg and effective rank.
>
> L2's effect on the Hessian is to add a positive definite (full-rank) constant to the original Hessian. Therefore, it is possible to choose a large enough L2 constant to ensure that every positive eigenvalue in the original Hessian stays positive in the regularized Hessian. We discuss this in **Corollary B.4**.
>
> - Missing related work: parseval regularization seems to capture a very similar same term, by orthogonalizing the weights.
> Thank you for pointing out this reference. **We have already added this to the manuscript**. We should note that Parseval regularization acts on the weight matrices where-as our L2-ER regularizer acts on the activations. These two methods are definitely related, but are distinct.
>
> > Why isnt the hessian rank measured in the experiments?
>
> We record the full Hessian spectrum; these plots can be found in figures 10, 14 and 15. We measure the degeneration of the Hessian spectrum with $\epsilon$-rank, where we measure the portion of absolute eigenvalues above $\epsilon$. In this way, we can ignore very small eigenvalues that contribute little to learning.

---

> ### Author Response · Authors · 2025-11-20
> **Rebuttal part 3**
>
> > - Doesn't this claim: "Single metrics are insufficient." undermine the narrative surrounding spectral collapse as the unifying explanation?
>
> Our work here aims to show how existing metrics can be generalized into the notion of spectral collapse. Through our framework, through analyzing the Hessian rank we are able to capture dead ReLUs, linearized ReLUs and co-linear gradients.  Our analysis unifies partial resetting methods like continual backpropogation [3] and spectral methods [2] by explaining how these methods affect the Hessian eigenspectrum.
>
> Current single metrics are insufficient to explain loss of plasticity. So we provided empirical evidence that spectral collapse is consistent with plasticity loss in all environments we tested, and we provided a theoretical framework to discuss current single metrics together under spectral collapse.
>
> > - Missing layer normalization baseline: it is now common to use normalization, and this has also been found to greatly reduce loss of plasticity.
>
> We have run layer-normalization on our permuted MNIST experiment, **updated in figure 3 (top left) and figure 7 (top)** and can confirm that it slightly loses plasticity even with L2 regularization. We are also running these baselines on the rest of our experiments. We will update the manuscript as the results come in.
>
> > - The neural network architectures here are rather small and basic, lacking normalization and skip connections. There is also comparatively little analysis of how the optimizer may contribute to curvature, as compared to previous work.
>
> Thank you for pointing this out. Our experiments are indeed **run on a variety of networks**. Permuted MNSIT is run with a small feedforward MLP (appendix H.1). Continual imagenet is run with a convolutional neural network (appendix I.1), and incremental cifar was run on a **resnet (appendix J.1) which includes skip connections**. We use sgd for superivised learning and adam for RL since they are two of the most popular optimizers used in the community and find that that both lead to loss of plasticity. Our existing analysis suggests that optimizers that improve the conditioning of the Hessian, like adamw through L2 regularization, would improve plasticity since it targets the neural newtwork curvature.
>
>
> Minor Comments
> > line 168:  is missing a subscript.
> Thank you, we have made the correction.
>
> Thank you again for taking the time to review the paper and providing helpful feedback! Do the above actions address your concerns with the paper? If not, what further clarification or modifications could we make to improve it?
>
>
>
> [1] Lewandowski, A., Tanaka, H., Schuurmans, D., & Machado, M. C. (2023, November 30). Curvature Explains Loss of Plasticity. http://arxiv.org/abs/2312.00246
>
> [2] Alex Lewandowski et al. (2024, June 10). Learning Continually by Spectral Regularization. http://arxiv.org/abs/2406.06811
>
> [3] Dohare, Shibhansh, et al. "Loss of plasticity in deep continual learning." Nature 632.8026 (2024): 768-774.

---

### Author Response · Authors · 2025-11-21
**Overall Rebuttal**

We appreciate the reviewers for their thoughtful comments on the paper. We would like to summarize the strengths and weaknesses of our submitted work to strengthen our contributions. **We would also like to emphasise that we have already integrated _all_ feedback into the manuscript, which appears in red in the current version in OpenReview**.

### Summary
Strength: All reviewers appreciate the theoretical analysis of $\tau$-trainability which connects dormant neurons, effective rank, and loss curvature to enrich the understanding of plasticity loss. This is indeed the central contribution of our paper.

Weakness: Reviewers broadly found experiments to be lacking a few baselines, including layernorm and spectralnorm. We have now finished running layer normalization and spectral regularization baselines on permuted MNIST. These results are already in the updated manuscript (Figure 3: top left, Figure 7 top). Layer normalisation with L2 still loses plasticity gradually. Spectral regularisation, while maintaining plasticity, runs much slower than our method due to power iteration at each gradient step (as shown in Figure 15). We are currently running these baselines on the rest of our experiments and will provide the results as the experiments finish.

Reviewers also found Figure 1 confusing. The figure is a toy construction meant to illustrate how ill-conditioned subspaces (flat “canyons”) versus well-conditioned basins (bowls) lead to different dynamics. In particular, initialisation in low-rank subspaces (canyons) can lead to poorer learning dynamics. We have updated the caption and also now point to the full description in Appendix C.

Contributions: Reviewer b6hD and reviewer c59C raised questions about the contributions of this project from the previous work [1, 2]. In particular, we take a curvature-based perspective on the problem like [1, 2], but we provide a formal theoretical framework on $\tau$-trainability that unifies weight norms, effective rank, and dormant neurons. We further study the full Hessian spectrum over various tasks, architectures, and algorithms to show the relationship graphs that demonstrate an alignment between spectral collapse and plasticity loss (Fig 2 and 7).

### Experiments added:
We have completed spectral regularization and layernorm on permuted MNIST. The figures have been updated (Figure 3: top left, Figure 7 top). We continue to run experiments and will update the manuscript as they are completed.


### Manuscript updates:
We have updated the introduction and related works. We have updated section 7 to better explain the KFAC approximation error and the derivation of our regularizer. **We have updates to the manuscript in red:**
- Included Parseval regularization [3]  in related work  as suggested by reviewer TdDm
- included Implicit regularisation [4]  in related work  as suggested by reviewer b6hD
- We have also promoted [1,2] to the introduction to help distinguish our contributions.
- We have adopted the changes suggested by TdDm to definitions 5.1 and 5.2.
- We applied the correction about trainability being in-terms of active neurons.
- We have updated section 7 with improved notation and added a derivation of the KFAC error to appendix D suggested by 4txZ.
- Updated figures 3, 6, 7 with new permuted MNIST experiments.
- Added figure 11, the Hessian spectrum at task end on permuted MNIST.
- Added figure 15, a runtime comparison of all algorithms on permuted MNIST.

We hope we have addressed the concerns of reviewers and will continue to provide updates as experiments roll in.

### References
[1] Lewandowski, Alex, et al. "Directions of curvature as an explanation for loss of plasticity." arXiv preprint arXiv:2312.00246 (2023).

[2] Lewandowski et al. (2024, June 10). Learning Continually by Spectral Regularization. http://arxiv.org/abs/2406.06811

[3] Chung, Wesley, et al. "Parseval regularization for continual reinforcement learning." Advances in Neural Information Processing Systems 37 (2024): 127937-127967.

[4] Kumar, Aviral, et al. "Implicit under-parameterization inhibits data-efficient deep reinforcement learning." arXiv preprint arXiv:2010.14498 (2020).

---

### Meta-Review · Area_Chair_EX9H · 2026-01-08

**Summary:**

The paper investigates the loss of plasticity in deep continual learning and find that this is due to Hessian spectral collapse. It further introduce \tau-trainability to unify existing plasticity-preserving approaches and proposes a curvature-regularized objective (L2-ER) to mitigate spectral collapse. Experiments on continual supervised and reinforcement learning tasks show that the proposed regularizer helps maintain plasticity and improves performance compared to standard baselines.

**Reviewer Concerns:**

Reviewers generally appreciated the theoretical analysis and proposed regularizers, but raised concerns on the lack of novelty on the L2-ER, incremental theorectical analysis, insufficient experimental comparisons, missing baselines (e.g., layer normalization, spectral regularization), limited connection between theory and experiments, lack of clarity in definitions and theorems, and some unclear writings requiring clarifications.

**Reviewer Scores:**

The reviewers scored 4, 2, 2, 4, tending to reject the submission.  The authors provided responses by comparing novelties with related works, adding new comparisons, and revising writings such as definitions, figures, etc. While these responses partially addressed the concerns, the novelty and theoretical contributions still appear incremental, and the new experimental evaluations are still insufficient. The submission cannot be accepted in the current version.

---

### Decision · Program_Chairs · 2026-01-26

Reject